ns# nature human behaviour

# Developmental changes in exploration resemble stochastic optimization

Anna P. Giron[1,2,12], Simon Ciranka [3,4,12], Eric Schulz [5], Wouter van den Bos[6,7], Azzurra Ruggeri [8,9,10], Björn Meder [8,11] & Charley M. Wu [1,3]✉

Human development is often described as a 'cooling off' process, analogous to stochastic optimization algorithms that implement a gradual reduction in randomness over time. Yet there is ambiguity in how to interpret this analogy, due to a lack of concrete empirical comparisons. Using data from *n* = 281 participants ages 5 to 55, we show that cooling off does not only apply to the single dimension of randomness. Rather, human development resembles an optimization process of multiple learning parameters, for example, reward generalization, uncertainty-directed exploration and random temperature. Rapid changes in parameters occur during childhood, but these changes plateau and converge to efficient values in adulthood. We show that while the developmental trajectory of human parameters is strikingly similar to several stochastic optimization algorithms, there are important differences in convergence. None of the optimization algorithms tested were able to discover reliably better regions of the strategy space than adult participants on this task.

Human development has fascinated researchers of both biological and artificial intelligence alike. As the only known process that reliably produces human-level intelligence[1], there is broad interest in characterizing the developmental trajectory of human learning[2–4] and understanding why we observe specific patterns of change[5].

One influential hypothesis describes human development as a 'cooling off' process[4,6,7], comparable to simulated annealing (SA)[8,9]. SA is a stochastic optimization algorithm named in analogy to a piece of metal that becomes harder to manipulate as it cools off. Initialized with high temperature, SA starts off highly flexible and likely to consider worse solutions as it explores the optimization landscape. But as the temperature cools down, the algorithm becomes increasingly greedy and more narrowly favours only local improvements, eventually converging on an (approximately) optimal solution. Algorithms with similar cooling mechanisms, such as stochastic gradient descent and its discrete

counterpart stochastic hill climbing (SHC) are abundant in machine learning and have played a pivotal role in the rise of deep learning[10–13].

This analogy of stochastic optimization applied to human development is quite alluring: young children start off highly stochastic and flexible in generating hypotheses[4,14–16] and selecting actions[17], which gradually tapers off over the lifespan. This allows children to catch information that adults overlook[18], and learn unusual causal relationships adults might never consider[4,15]. Yet this high variability also results in large deviations from reward-maximizing behaviour[3,19–22], with gradual improvements during development. Adults, in turn, are well calibrated to their environment and quickly solve familiar problems, but at the cost of flexibility, since they experience difficulty adapting to novel circumstances[23–26].

While intuitively appealing, the implications and possible boundaries of the optimization analogy remain ambiguous without a clear

[1]Human and Machine Cognition Lab, University of Tübingen, Tübingen, Germany. [2]Attention and Affect Lab, University of Tübingen, Tübingen, Germany. [3]Center for Adaptive Rationality, Max Planck Institute for Human Development, Berlin, Germany. [4]Max Planck UCL Centre for Computational Psychiatry and Ageing Research, Berlin, Germany. [5]MPRG Computational Principles of Intelligence, Max Planck Institute for Biological Cybernetics, Tübingen, Germany. [6]Department of Psychology, University of Amsterdam, Amsterdam, the Netherlands. [7]Amsterdam Brain and Cognition, University of Amsterdam, Amsterdam, the Netherlands. [8]MPRG iSearch, Max Planck Institute for Human Development, Berlin, Germany. [9]School of Social Sciences and Technology, Technical University Munich, Munich, Germany. [10]Central European University, Vienna, Austria. [11]Institute for Mind, Brain and Behavior, Health and Medical University, Potsdam, Germany. [12]These authors contributed equally: Anna P. Giron, Simon Ciranka. ✉e-mail: charley.wu@uni-tuebingen.de

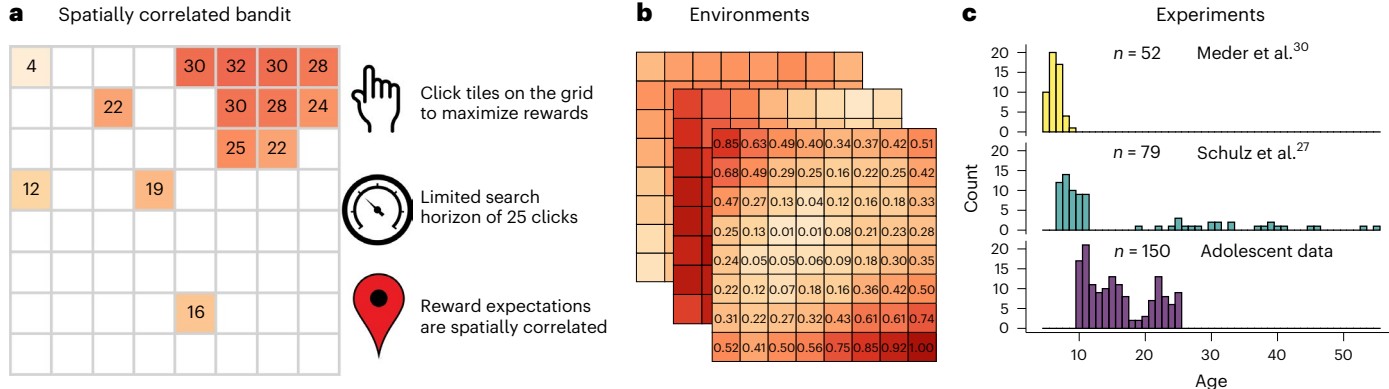

**Fig. 1 | Experiments and model overview. a**, Illustration of the task. **b**, Fully revealed examples of the spatially correlated reward environments (showing expected reward), where nearby tiles tended to have similar rewards. **c**, Description of the three experiments, showing the number of participants and their age distribution (after filtering; Methods).

definition of the process and a specification of what is being optimized. As a consequence, there is a need for a direct empirical test of the similarities and differences between human development and algorithmic optimization.

Perhaps the most direct interpretation is to apply cooling off to the single dimension of random decision temperature, controlling the amount of noise when selecting actions or sampling hypotheses[6,16,27], although alternative implementations are also possible[28,29]. Evidence from experimental studies suggest that young children are harder to predict than adults[17,30], implying greater stochasticity, which is amplified in neurodevelopmental disorders such as attention deficit hyperactivity disorder[28] and impulsivity[31]. However, this interpretation is only part of the story, since developmental differences in choice variability can be traced to changes affecting multiple aspects of learning and choice behaviour. Aside from a decrease in randomness, development is also related to changes in more systematic, uncertainty-directed exploration[19,27,32], which is also reduced over the lifespan. Additionally, changes in how people generalize rewards to novel choices[27] and the integration of new experiences[33,34] affect how beliefs are formed and different actions are valued, also influencing choice variability. In sum, while decision noise certainly diminishes over the lifespan, this is only a single aspect of human development.

Alternatively, one could apply the cooling off metaphor to an optimization process in the space of learning strategies, which can be characterized across multiple dimensions of learning. Development might thus be framed as parameter optimization, which tunes the parameters of an individual's learning strategy, starting off by making large tweaks in childhood, followed by gradually lesser and more-refined adjustments over the lifespan. In the stochastic optimization metaphor, training iterations of the algorithm become a proxy for age.

This interpretation connects the metaphor of stochastic optimization with Bayesian models of cognitive development, which share a common notion of gradual convergence[35,36]. In Bayesian models of development, individuals in early developmental stages possess broad priors and vague theories about the world, which become refined with experience[35]. Bayesian principles dictate that, over time, novel experiences will have a lesser impact on future beliefs or behaviour as one's priors become more narrow[36,37]. Observed over the lifespan, this process will result in large changes to beliefs and behaviour early in childhood and smaller changes in later stages, implying a similar developmental pattern as the stochastic optimization metaphor. In sum, not only might the outcomes of behaviour be more stochastic during childhood, but the changes to the parameters governing behaviour might also be more stochastic in earlier developmental stages.

## Goals and scope

In this work, we aim to resolve ambiguities around commonly used analogies to stochastic optimization in developmental psychology. While past work has compared differences in parameters between discrete age groups in both structured[27,30] and unstructured reward domains[19,32,33], here we characterize the shape of developmental change across the lifespan, from ages 5 to 55. Instead of relying on verbal descriptions, we use formal computational models to clarify which cognitive processes are being tweaked during development through explicit commitments to free parameters. We then directly compare the trajectory of various optimization algorithms to age differences in those parameters, allowing us to finally put the metaphor to a direct empirical test.

Behavioural analyses show that rather than a uni-dimensional transition from exploration to exploitation, human development produces improvements in both faculties. Through computational models, we find simultaneous changes across multiple dimensions of learning, starting with large tweaks during childhood and plateauing in adulthood. We then provide direct empirical comparisons to multiple optimization algorithms as a meta-level analysis to describe changes in model parameters over the lifespan, where the best-performing algorithm is the most similar to human development. However, we also find notable differences in convergence between human development and algorithmic optimization. Yet, this disparity fails to translate into reliable differences in performance, suggesting a remarkable efficiency of human development.

## Results

We analyse experimental data from $n = 281$ participants between the ages of 5 and 55, performing a spatially correlated multi-armed bandit task that is both intuitive and richly complex (Fig. 1a)[38]. Participants were given a limited search horizon (25 choices) to maximize rewards by either selecting an unobserved or previously revealed tile on an $8 \times 8$ grid. Each choice yielded normally distributed rewards, with reward expectations correlated based on spatial proximity (Fig. 1b), such that tiles close to one another tended to have similar rewards. Since the search horizon was substantially smaller than the number of unique options, generalization and efficient exploration were required to obtain high rewards.

Our dataset (Fig. 1c) combines openly available data from two previously published experiments[27,30] targeting children and adult participants ($n = 52$ and $n = 79$, respectively; after filtering), along with new unpublished data ($n = 150$) targeting the missing gap of adolescents. Although experimental designs differed in minor details (for example, tablet versus computer; Methods), the majority of differences were removed by filtering out participants (for example, assigned to a

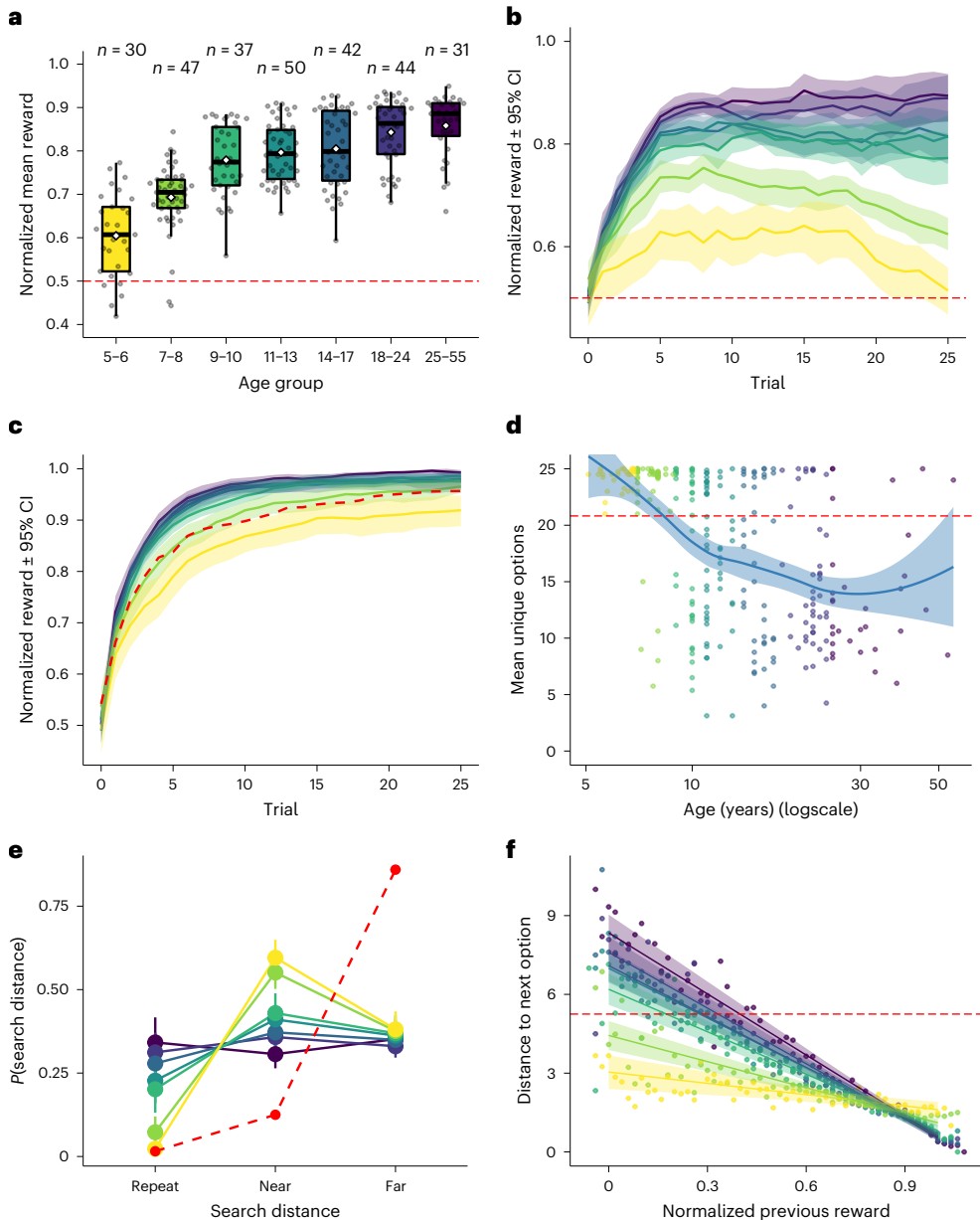

**Fig. 2 | Behavioural results. a**, Mean reward across age groups. Each dot is one participant, Tukey boxplots showing median and 1.5 × interquartile range (IQR), with the white diamond indicating the group mean. The red dashed line indicates a random baseline (in all plots). The same colours are used to indicate age in all plots. **b**, Learning curves showing mean reward over trials, averaged across all rounds. Lines indicate group means, while the ribbons show the 95% CI. **c**, Learning curves showing maximum reward earned up until a given trial, averaged across all rounds. **d**, The number of unique options sampled per round

as a function of age. Each dot is one participant, while the line and ribbon show a locally smoothed regression (±95% CI). **e**, The proportion of repeat, near (distance = 1) and far (distance > 1) choices as a function of age. Each dot indicates a group mean, while the error bars indicate the 95% CI. **f**, Search distance as a function of the previous reward value. Each line is the fixed effect of a hierarchical Bayesian regression (Supplementary Table 3) with the ribbons indicating 95% CI. Each dot is the mean of the raw data.

different class of reward environments). Reliability tests revealed no differences in performance, model accuracy and parameter estimates for overlapping age groups across experiments (all $P > 0.128$ and Bayes Factor (BF) < 0.73; Supplementary Fig. 1 and Supplementary Table 1). Additionally, robust model and parameter recovery (Supplementary Figs. 4–6) provide high confidence in our ability to capture the key components of learning across the lifespan.

## Behavioural analyses

We first analysed participant performance and behavioural patterns of choices (Fig. 2). We treat age as a continuous variable when possible,

but also discretize participants into seven similarly sized age groups ($n \in [30, 50]$, see Supplementary Table 4 for exact sample sizes). These behavioural results reveal clear age-related trends in learning and exploration captured by our task.

**Performance.** Participants monotonically achieved higher rewards as a function of age (Pearson correlation: $r = 0.51$, $P < 0.001$, BF > 100), with even the youngest age group (five-to-six year olds) strongly outperforming chance (one-sample $t$ test: $t(29) = 5.1$, $P < 0.001$, Cohen's $d = 0.9$, BF > 100; Fig. 2a). The learning curves in Fig. 2b show average reward as a function of trial, revealing a similar trend, with older

participants displaying steeper increases in average reward. Notably, the two youngest age groups (five-to-six and seven-to-eight year olds) displayed decaying learning curves with decreasing average reward on later trials, suggesting a tendency to overexplore (supported by subsequent analyses below). We did not find any reliable effect of learning over rounds (Supplementary Fig. 2).

We also analysed maximum reward (up until a given trial) as a measure of exploration efficacy, where older participants reliably discovered greater maximum rewards (Kendall rank correlation: $r_\tau = 0.23$, $P < 0.001$, BF > 100) and showed steeper increases on a trial-by-trial basis (Fig. 2c). Thus, the reduction in the average reward acquired by the youngest age groups did not convert into improved exploration outcomes, measured in terms of maximum reward.

**Behavioural patterns.** Next, we looked at search patterns to better understand the behavioural signatures of age-related changes in exploration. The youngest participants (five-to-six year olds) sampled more unique options than chance (Wilcoxon signed-rank test: $Z = -4.7$, $P < 0.001$, $r = -0.85$, BF > 100), but also less than the upper-bound on exploration (that is, unique options on all 25 trials: $Z = -4.1$, $P < 0.001$, $r = -0.76$, BF > 100). The number of unique options decreased strongly as a function of age ($r_\tau = -0.33$, $P < 0.001$, BF > 100; Fig. 2d), consistent with the overall pattern of reduced exploration over the lifespan. Note that all participants were informed and tested about the fact they could repeat choices.

We then classified choices into repeat, near (distance = 1) and far (distance > 1), and compared this pattern of choices to a random baseline (red dashed line; Fig. 2e). Five-to-six year olds started off with very few repeat choices (comparable to chance: $Z = 0.9$, $P = 0.820$, $r = 0.17$, BF = 0.27) and a strong preference for near choices (more than chance: $Z = -4.6$, $P < 0.001$, $r = -0.84$, BF > 100). Over the lifespan, the rate of repeat choices increased, while near decisions decreased, gradually reaching parity for 14–17 year olds (comparing repeat versus near: $Z = -1.0$, $P = 0.167$, $r = -0.15$, BF = 0.46) and remaining equivalent for all older age groups (all $P > 0.484$, BF < 0.23). In contrast, the proportion of far choices remained unchanged over the lifespan ($r_\tau = -0.08$, $P = 0.062$, BF = 0.48). These choice patterns indicate that even young children do not simply behave randomly, with the amount of randomness decreasing over time. Rather, younger participants exploit past options less than older participants (repeat choices), preferring instead to explore unknown tiles within a local radius (near choices). While the tendency to prefer exploring near rather than far options gradually diminished over the lifespan, this preference for local search distinguished participants of all age groups from the random model.

Lastly, we analysed how reward outcomes influenced search distance using a Bayesian hierarchical regression (Fig. 2f and Supplementary Table 3). This model predicted search distance as a function of the previous reward value and age group (including their interaction), with participants treated as random effects. This can be interpreted as a continuous analogue to past work using a win–stay lose–shift strategy[17,38], and provides initial behavioural evidence for reward generalization. We found a negative linear relationship in all age groups, with participants searching locally following high rewards and searching further away after low rewards. This trend becomes stronger over the lifespan, with monotonically more negative slopes over the lifespan (Supplementary Table 3). While all age groups adapted their search patterns in response to reward, the degree of adaptation increased over the lifespan.

**Behavioural summary.** To summarize, younger children tended to explore unobserved tiles instead of exploiting options known to have good outcomes. This can be characterized as overexploration since increased exploration did not translate into higher maximum rewards (Fig. 2c). Older participants explored less but more effectively, and were more responsive in adapting their search patterns to reward observations (Fig. 2f). We now turn to model-based analyses to complement

these results with a more precise characterization of how the different mechanisms of learning and exploration change over the lifespan.

## Model-based analyses

We conducted a series of reinforcement learning[39] analyses to characterize changes in learning over the lifespan. We first compared models in their ability to predict out-of-sample choices (Fig. 3b,c) and simulate human-like learning curves across age groups (Fig. 3d). We then analysed the parameters of the winning model (Fig. 3e,f), which combined Gaussian process (GP) regression with upper confidence bound (UCB) sampling (described below). These parameters allow us to describe how three different dimensions of learning change with age: generalization ($\lambda$; equation (2)), uncertainty-directed exploration ($\beta$; equation (3)) and decision temperature ($\tau$; equation (4)). We then compare the developmental trajectory of these parameters to different stochastic optimization algorithms (Fig. 4).

**Modelling learning and exploration.** We first describe the GP-UCB model (Fig. 3a) combining all three components of generalization, exploration and decision temperature. We then lesion away each component to demonstrate all are necessary for describing behaviour (Fig. 3b,c). We describe the key concepts below, while Fig. 3a provides a visual illustration of the model (see Methods for details).

GP regression[40] provides a reinforcement learning model of value generalization[38], where past reward observations can be generalized to novel choices. Here, we describe generalization as a function of spatial location, where closer observations exhibit a larger influence. However, the same model can also be used to generalize based on the similarity of arbitrary features[41] or based on graph-structured relationships[42].

Given previously observed data $\mathcal{D}_t = \{\mathbf{X}_t, \mathbf{y}_t\}$ of choices $\mathbf{X}_t = [\mathbf{x}_1, \ldots, \mathbf{x}_t]$ and rewards $\mathbf{y}_t = [y_1, \ldots, y_t]$ at time $t$, the GP uses Bayesian principles to compute posterior predictions about the expected rewards $r_t$ for any option $\mathbf{x}$:

$$p(r_t(\mathbf{x})|\mathcal{D}_t) \sim \mathcal{N}(m_t(\mathbf{x}), v_t(\mathbf{x})). \tag{1}$$

The posterior in equation (1) takes the form of a Gaussian distribution, allowing it to be fully characterized by posterior mean $m_t(\mathbf{x})$ and uncertainty $v_t(\mathbf{x})$ (that is, variance; see equations (6) and (7) for details and Fig. 3a for an illustration).

The posterior mean and uncertainty predictions critically depend on the choice of kernel function $k(\mathbf{x}, \mathbf{x}')$, for which we use a radial basis function describing how observations from one option $\mathbf{x}$ generalize to another option $\mathbf{x}'$ as a function of their distance:

$$k(\mathbf{x}, \mathbf{x}') = \exp\left(-\frac{||\mathbf{x} - \mathbf{x}'||^2}{2\lambda^2}\right). \tag{2}$$

The model thus assumes that nearby options generate similar rewards, with the level of similarity decaying exponentially over increased distances. The generalization parameter $\lambda$ describes the rate at which generalization decays, with larger estimates corresponding to stronger generalization over greater distances.

We then use UCB sampling to describe the value of each option $q(\mathbf{x})$ as a weighted sum of expected reward $m(\mathbf{x})$ and uncertainty $v(\mathbf{x})$:

$$q(\mathbf{x}) = m(\mathbf{x}) + \beta\sqrt{v(\mathbf{x})}. \tag{3}$$

$\beta$ captures uncertainty-directed exploration, determining the extent that uncertainty is valued positively, relative to exploiting options with high expectations of reward.

Lastly, we use a softmax policy to translate value $q(\mathbf{x})$ into choice probabilities:

$$p(\mathbf{x}) \propto \exp(q(\mathbf{x})/\tau). \tag{4}$$

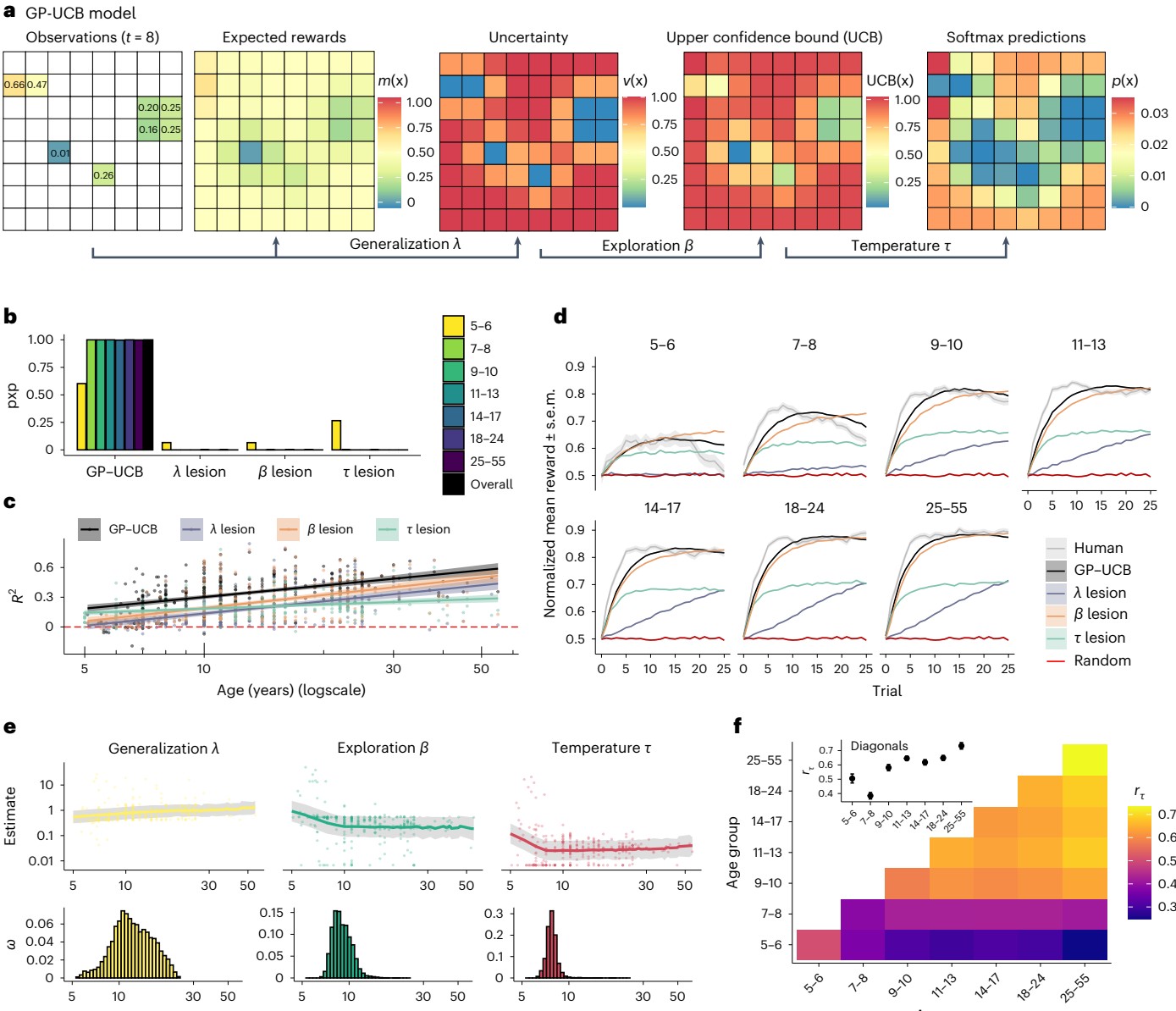

**Fig. 3 | Model results. a**, Illustration of the GP-UCB model based on observations at the eighth trial (showing normalized reward). We use GP regression as a psychological model of reward generalization[38], making Bayesian estimates about the expected rewards and uncertainty for each option. The free parameter $\lambda$ (equation (2)) controls the extent that past observations generalize to new options. The expected rewards $m(\mathbf{x})$ and uncertainty estimates $v(\mathbf{x})$ are combined using UCB sampling (equation (3)) to produce a valuation for each option. The exploration bonus $\beta$ governs the value of exploring uncertain options relative to exploiting high reward expectations. Lastly, UCB values are entered into a softmax function (equation (4)) to make probabilistic predictions about where the participant will search next. The decision temperature parameter $\tau$ governs the amount of random (undirected) exploration. **b**, Hierarchical Bayesian model selection, where pxp defines the probability of each model being predominant

in the population (see Supplementary Fig. 2 and Supplementary Table 3 for more details). **c**, Predictive accuracy ($R^2$) as a function of age. Each dot is a participant and the lines and ribbons show the slope (±95% CI) of a linear regression. **d**, Simulated learning curves, using participant parameter estimates. Human data illustrates the mean (±95% CI), while model simulations report the mean. **e**, Top, participant parameter estimates as a function of age. Each dot is a single participant, with the line and ribbon showing the posterior predictions (±95% CI) from a Bayesian changepoint regression model. Bottom, posterior distribution of which age the changepoint ($\omega$) is estimated to occur. **f**, Similarity matrix of parameter estimates. Using Kendall's rank correlation ($r_\tau$), we report the similarity of parameter estimates both within and between age groups. The within age group similarities (diagonals) are also visualized in the inset plot, where dots show the mean and error bars indicate the 95% CI. Refer to Fig. 2a for sample sizes of age groups.

---

The decision temperature $\tau$ controls the amount of random exploration. Larger values for $\tau$ introduce more choice stochasticity, where $\tau \to \infty$ converges on a random policy.

**Lesioned models.** To ensure all components of the GP-UCB model play a necessary role in capturing behaviour, we created model variants lesioning away each component. The $\lambda$ lesion model removed

the capacity for generalization, by replacing the GP component with a Bayesian reinforcement learning model that assumes independent reward distributions for each option (equations (8) to (11)). The $\beta$ lesion model removed the capacity for uncertainty-directed exploration by fixing $\beta = 0$, thus valuing options solely based on expectations of reward $m(\mathbf{x})$. Lastly, the $\tau$ lesion model swapped the softmax policy for an epsilon-greedy policy, as an alternative form of choice

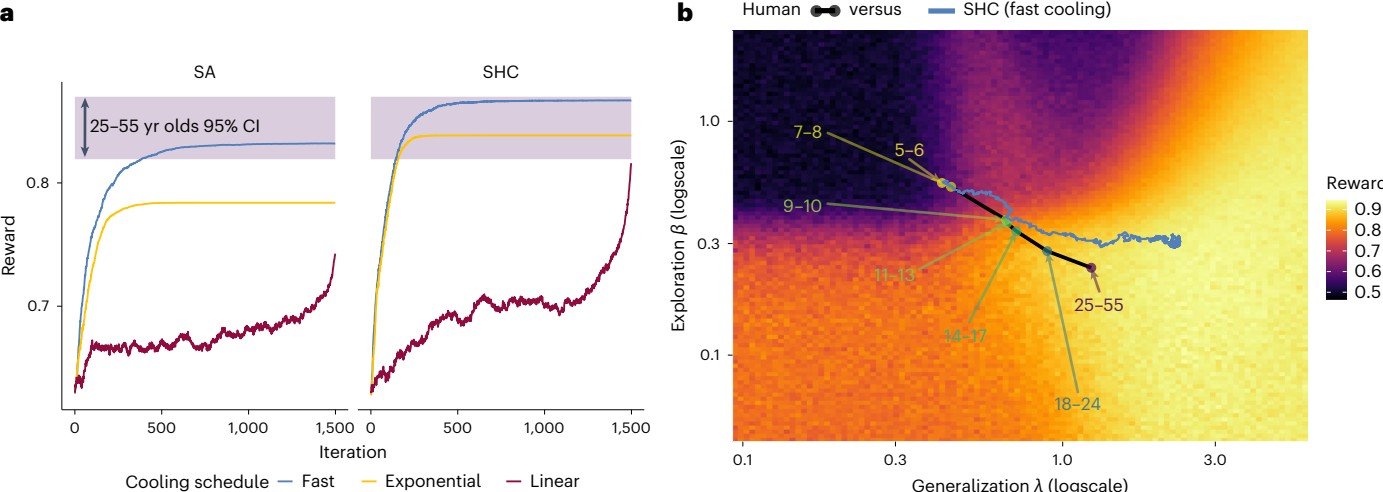

**Fig. 4 | Developmental trajectory. a**, Mean reward in each iteration for SA and SHC algorithms, combined with fast, exponential and linear cooling schedules. The shaded purple band indicates the 95% CI of human performance in the 25–55 age group. **b**, Comparison of human and algorithm trajectories, focusing on the best-performing SHC fast cooling algorithm. The annotated dots show the median parameter estimates of each age group, while the blue line shows the trajectory of the optimization algorithm (median overall all simulations). This two-dimensional illustration focuses on changes in generalization $\lambda$ and directed exploration $\beta$ (see Supplementary Fig. 6 for all algorithms and all parameter comparisons), with the underlying fitness landscape depicted using the median $\tau$ estimate across all human data ($\bar{\tau} = 0.03$).

stochasticity: with probability $p(\epsilon)$ a random option is sampled, and with probability $p(1 - \epsilon)$ the option with the highest UCB value is sampled (equation (12)).

**Model comparison.** We fitted all models using leave-one-round-out cross-validation (Methods). We then conducted hierarchical Bayesian model selection[43] to compute the protected exceedance probability (pxp) for describing which model is most likely in the population. We found that GP-UCB was the best model for each individual age group and also aggregated across all data (Fig. 3a). There is still some ambiguity between models in the five-to-six year old group, but this quickly disappears in all subsequent age groups ($pxp_{GP\text{-}UCB} > 0.99$). Figure 3b describes the out-of-sample predictive accuracy of each model as a continuous function of age, where a pseudo-$R^2$ provides an intuitive comparison to random chance (equation (13)). Intuitively, $R^2 = 0$ indicates chance-level predictions and $R^2 = 1$ indicates theoretically perfect predictions. While there is again some ambiguity among five-to-six year olds, GP-UCB quickly dominates and remains the best model across all later ages.

Aside from only predicting choices, we also simulated learning curves for each model (using median participant parameter estimates) and compared them against human performance for each age group (Fig. 3c). The full GP-UCB model provides the best description across all age groups, although the $\beta$ lesion model also produces similar patterns. However, only GP-UCB by virtue of the exploration bonus $\beta$ is able to recreate the decaying learning curves for five-to-six and seven-to-eight year olds. Altogether, these results reveal that all three components of generalization ($\lambda$), uncertainty-directed exploration ($\beta$) and decision temperature ($\tau$) play a vital role in describing behaviour. Next, we analyse how each of these parameters changes over the lifespan.

**Parameters.** We use both regression and similarity analyses to interpret age-related changes in GP-UCB parameters. The regression (Fig. 3e) modelled age-related changes in the log-transformed parameters using a multivariate Bayesian changepoint regression[44] (Methods). This approach models the relationship between age and each parameter as separate linear functions separated by an estimated changepoint $\omega$, at which point the regression slope changes from $b_1$ to

$b_2$ (equation (15)). Using leave-one-out cross-validation, we established that this simple changepoint model predicted all GP-UCB parameters better than linear or complex regression models up to fourth-degree polynomials (both with and without log-transformed variables and compared against lesioned intercept-only variants; Supplementary Tables 5 and 6).

The regression analysis (Fig. 3e) revealed how all parameters changed rapidly during childhood (all $b_1$ confidence intervals (CIs) different from 0), but then plateaued such that there were no credible changes in parameters after the estimated changepoint (all $b_2$ CIs overlapped with 0; Supplementary Table 7). More specifically, generalization increased ($b_1(\lambda) = 0.08 [0.02, 0.26]$) until around 13 years of age ($\omega(\lambda) = 12.7 [7.70, 19.80]$), whereas there were sharp decreases in directed exploration ($b_1(\beta) = -0.39 [-0.79, -0.13]$) and decision temperature ($b_1(\tau) = -0.59 [-1.05, -0.25]$), until around nine ($\omega(\beta) = 9.10 [7.44, 11.40]$) and eight ($\omega(\tau) = 7.74 [6.88, 8.77]$) years of age, respectively. In a multivariate similarity analysis, we computed the pair-wise similarity of all parameter estimates between participants (Kendall's $\tau$), which we then averaged over age groups (Fig. 3f). This shows that older participants were more similar to each other than were younger participants (Fig. 3f inset), suggesting development produces a convergence towards a more similar set of parameters. Whereas older participants achieved high rewards using similar learning strategies, younger participants tended to overexplore and acquired lower rewards, but each in their own fashion, with more-diverse strategies.

These results highlight how development produces changes to all parameters governing learning, not only a uni-dimensional reduction in random sampling. An initially steep but plateauing rate of change across model parameters is broadly consistent with the metaphor of stochastic optimization in the space of learning strategies (Fig. 3e). The increasing similarity in participants' parameters again speaks for a developmental process that gradually converges on a configuration of learning parameters (Fig. 3f), which can also be used to generate better performance (Fig. 3d).

**Comparison of human and algorithmic trajectories**
Beyond qualitative analogies, we now present a direct empirical comparison between human development and stochastic optimization.

We first computed a fitness landscape (Supplementary Fig. 7) across one million combinations of plausible parameter values of the GP-UCB model (Methods), with each parameter combination yielding a mean reward based on 100 simulated rounds. We then simulated different optimization algorithms on the fitness landscape, using each of the cross-validated parameter estimates of the five-to-six year old age group as initialization points. Specifically, we tested SA and SHC in combination with three common cooling schedules (fast cooling, exponential cooling and linear cooling; Methods). Intuitively, SA uses rejection sampling to preferentially select better solutions in the fitness landscape (equation (16)), with higher optimization temperatures relaxing this preference. SHC is a discrete analogue of stochastic gradient descent, selecting new solutions proportional to their relative fitness (equation (17)), with higher optimization temperatures making it more likely to select lower fitness solutions.

While we do not attempt to curve-fit the exact cooling schedule that best describes human development, we observe that fast and exponential cooling generally performed better than linear cooling (Fig. 4a). The metaphor therefore holds: we do not observe linear changes in development, but rather rapid initial changes during childhood, followed by a gradual plateau and convergence. Yet remarkably, neither SA nor SHC converged on reliably better solutions than adult participants aged 25–55 (SA fast cooling $t(149) = -0.4$, $P = 0.675$, $d = 0.08$, BF = 0.23; SHC fast cooling $t(149) = 0.8$, $P = 0.447$, $d = 0.2$, BF = 0.27).

Figure 4b compares the developmental trajectory of human participants (labelled dots) to the trajectory of the best-performing SHC (fast cooling) algorithm (blue line; see Supplementary Fig. 8 for all algorithms and all parameter comparisons and Supplementary Fig. 9 for variability of trajectories). We focus on changes in generalization and exploration parameters since rewards decrease monotonically with increased decision temperature. Particularly for younger age groups, age-related changes in parameters follow a similar trajectory to the optimization algorithms. However, a notable divergence emerges around adolescence (ages 14–17).

To concretely relate human and algorithmic trajectories, we estimated the same changepoint regression model (Fig. 3e) on the sequence of algorithmic parameters from the SHC (fast cooling) algorithm (see Supplementary Fig. 10 for details). This revealed a similar pattern of rapid change before the changepoint with all $b_1$ slopes having the same direction as humans, followed by a plateau of $b_2$ slopes around 0 (Supplementary Fig. 10a and Supplementary Table 8).

This analysis also allows us to quantify convergence differences between human development and stochastic optimization. For a given dataset (human versus algorithm) and a given parameter ($\lambda, \beta, \tau$), we use the respective upper 95% CI of $\omega$ estimates as a threshold for convergence (Supplementary Fig. 10b). Comparing a matched sample of parameter estimates after the convergence threshold, we find that human generalization $\lambda$ and directed-exploration $\beta$ parameters converged at lower values than the algorithm ($\lambda$: $U = 2740$, $P < 0.001$, $r_\tau = -0.38$, BF > 100; $\beta$: $U = 10,114$, $P < 0.001$, $r_\tau = -0.19$, BF > 100), while decision temperature $\tau$ was not credibly higher or lower ($U = 22,377$, $P = 0.188$, $r_\tau = 0.05$, BF = 0.12; Supplementary Fig. 10c).

Since these deviations nevertheless fail to translate into reliable differences in performance, this may point towards resource rational constraints on human development[45,46]: aside from only optimizing for the best performance, the cognitive costs of different strategies may also be considered (Discussion).

## Discussion

From a rich dataset of $n = 281$ participants ages 5 to 55, our results reveal human development cools off not only in the search for rewards or hypotheses, but also in the search for the best learning strategy. Thus, the stochastic optimization metaphor best applies to a multivariate optimization of all parameters of a learning model, rather than only

a decrease in randomness when selecting actions or hypotheses. What begins as large tweaks to the cognitive mechanisms of learning and exploration during childhood gradually plateaus and converges in adulthood (Fig. 3e,f). This process is remarkably effective, resembling the trajectory of the best-performing stochastic optimization algorithm (SHC fast cooling; Fig. 4) as it optimizes the psychologically interpretable parameters of a Bayesian reinforcement learning model (GP-UCB). While there are notable differences in the solutions that human development and stochastic optimization converged upon, none of the algorithms achieved reliably better performance than adult human participants (25–55 year olds; Fig. 4a). This work provides important insights into the nature of developmental changes in learning and offers normative explanations for why we observe these specific developmental patterns.

Rather than a uni-dimensional transition from exploration to exploitation over the lifespan[25], we observe refinements in both the ability to explore (Fig. 2c) and exploit (Fig. 2f), with monotonic improvements in both measures as a function of age. While even five-to-six year olds perform better than chance (Fig. 2a), exploration becomes more effective over the lifespan, as larger reward values are discovered despite sampling fewer unique options (Fig. 2c,d). Meanwhile, exploitation becomes more responsive, with older participants adapting their search distance more strongly based on reward outcomes (Fig. 2e). This resembles a developmental refinement of a continuous win–stay lose–shift heuristic[17,38], and is consistent with the hypothesis that people use heuristics more efficiently as they age[47]. These results also reaffirm past work showing a reduction of stochasticity over the lifespan[4,14,16], but expand the scope of developmental changes across multiple dimensions of learning.

With a reinforcement learning model (GP-UCB), we characterize age-related changes in learning through the dimensions of generalization ($\lambda$), uncertainty-directed exploration ($\beta$) and decision temperature ($\tau$). All three dimensions play an essential role in predicting choices (Fig. 3b,c) and simulating realistic learning curves across all ages (Fig. 3d), with recoverable models and parameter estimates (Supplementary Figs. 4–6).

Changes in all parameters occur rapidly during childhood (increase in generalization and decrease in both exploration and decision temperature), but then plateau around adolescence (Fig. 3e). Younger participants tend to be more diverse, whereas adults are more similar to one another, with continued convergence of parameter estimates until adulthood (25–55 year olds; Fig. 3f). Both the reduction of age-related differences and the increasing similarity of parameters support the analogy of development as a stochastic optimization process, which gradually converges upon an (approximately optimal) configuration of learning parameters.

Our direct comparison between the developmental trajectory of human parameters and various stochastic optimization algorithms (Fig. 4) revealed both striking similarities and intriguing differences. The best-performing algorithm (SHC with fast cooling) also most resembles the parameter trajectory of human development (Supplementary Fig. 8), suggesting that optimization provides a useful characterization of developmental changes in learning strategy.

However, there are also limitations to the metaphor, and it should be noted that different parameters are optimal in different contexts[48,49]. Other developmental studies using reinforcement learning models suggest older participants may display more-optimal parameters in general[3], by being better able to adapt their strategies to task demands. This raises the question of whether children indeed have less optimal parameters per se or are simply slower when adapting to the task. We partially address this possibility by analysing performance over rounds, where we found no reliable age-related differences in learning over rounds (Supplementary Fig. 2). Thus, it seems unlikely that given more time to adapt to the task (for example, adding more rounds), children would register more optimal parameters.

Nevertheless, suboptimality of task-specific parameters does not suggest that younger individuals are maladaptive from a developmental context. Rather, development prepares children to learn about the world more generally, beyond the scope of any specific experimental paradigm or computational model. In line with the stochastic optimization analogy, younger children could try diverse strategies that our model does not account for, which then registers as suboptimal parameter estimates. This is consistent with the result that the predictive accuracy of the model generally increases over the lifespan (that is, $R^2$; Fig. 3c).

We also observed intriguing differences in the parameters that humans converged on compared to the algorithm trajectories, with adult participants displaying lower generalization and less uncertainty-directed exploration (Supplementary Fig. 10). Yet remarkably, none of the optimization algorithms achieved significantly better performance than adult participants (Fig. 4). Thus, these differences might point towards cognitive costs, which are not justified by any increased performance benefits. Generalization over a greater extent may require remembering and performing computations over a larger set of past observations[42,50], which is why some GP approximations reduce the number of inputs to save computational costs[51]. Similarly, deploying uncertainty-directed exploration is also associated with increased cognitive costs[52], and can be systematically diminished through working memory load[53] or time pressure[54] manipulations.

### Limitations and future directions

One limitation of our analyses is that we rely on cross-sectional rather than longitudinal data, observing changes in learning not only across the lifespan but also across individuals. Yet despite the advantages of the longitudinal study, it might not be appropriate in this setting because we would be unable to distinguish between performance improvement due to cognitive development and practice. Having participants repeatedly interact with the same task at different stages of development could conflate task-specific changes in reward learning with domain-general changes in their learning strategy. Yet future longitudinal analyses may be possible using a richer paradigm. For instance, modelling how we learn intuitive theories about the world[55] or compositional programs[56] as a search process in some latent hypothesis space. The richness of these domains may allow similar dimensions of learning to be measured from sufficiently distinct tasks administered at different developmental stages.

While we have characterized behavioural changes in learning using the distinct and recoverable parameters of a reinforcement learning model, future work is needed to relate these parameters to the development of specific neural mechanisms. Existing research provides some promising candidates. Blocking dopamine D2 receptors has been shown to impact stimulus generalization[57], selectively modulating similarity-based responses in the hippocampus. Similar multi-armed bandit tasks have linked the frontopolar cortex and the intraparietal sulcus to exploratory decisions[58], where more specifically the right frontopolar cortex has been causally linked to uncertainty-directed exploration[59], which can be selectively inhibited via transcranial magnetic stimulation.

Stochastic optimization also allows for 're-heating'[8] by adding more flexibility in later optimization stages. Re-heating is often used in dynamic environments or when insensitivities of the fitness landscape can cause the algorithm to get stuck. Since deviations from the algorithm trajectories start in adolescence, this may coincide with a second window of developmental plasticity during adolescence[60,61]. While we observed relatively minor changes in the parameters governing individual learning, plasticity in adolescents is thought to specifically target social learning mechanisms[7,62]. Thus, different aspects of development may fall under different cooling schedules and similar analyses should also be applied to other learning contexts.

Additionally, our participant sample is potentially limited by a WEIRD (western, educated, industrialized, rich and democratic) bias[63], where the relative safety of developmental environments may promote more exploration. While we would expect to find a similar qualitative pattern in more-diverse cultural settings, different expectations about the richness or predictability of environments may promote quantitatively different levels of each parameter. Indeed, previous work using a similar task but with risky outcomes found evidence for a similar generalization mechanism, but a different exploration strategy that prioritized safety[64].

Finally, our research also has implications for the role of the environment in maladaptive development. Our comparison to stochastic optimization suggests, in line with life history theory[65] and empirical work in rodents and humans[4,61,66], that childhood and adolescence are sensitive periods for configuring learning and exploration parameters. Indeed, adverse childhood experiences have been shown to reduce exploration and impair reward learning[67]. Organisms utilize early life experience to configure strategies for interacting with their environment, which for most species remain stable throughout the lifespan[68]. Once the configuration of learning strategies has cooled off, there is less flexibility for adapting to novel circumstances in later developmental stages. In the machine learning analogy we have used, some childhood experiences can produce a mismatch between training and test environments, where deviations from the expected environment have been linked to a number of psychopathologies[69]. Such a mismatch would set the developmental trajectory towards regions of the parameter space that are poorly suited for some features of the adult environment, but may provide hidden benefits for other types of problems more similar to the ones encountered during development[70]. Rather than only focusing on adult phenotypes at a single point in time, accounting for adaptation and optimization over the lifespan provides a more complete understanding of developmental processes.

### Conclusions

Scientists often look to statistical and computational tools for explanations and analogies[71]. With recent advances in machine learning and artificial intelligence, these tools are increasingly vivid mirrors into the nature of human cognition and its development. We can understand idiosyncrasies of hypothesis generation through Monte Carlo sampling[72], individual learning through optimization[73–75], and development as programming or 'hacking'[56]. An important advantage of computational explanations is that they offer direct empirical demonstrations, instead of remaining as vague, verbal comparisons. Here, we provided such a demonstration, and added much-needed clarity to commonly used analogies of stochastic optimization in developmental psychology.

## Methods

### Experiments

All experiments were approved by the ethics committee of the Max Planck Institute for Human Development (protocols ABC2016/08, ABC2017/04 and A2018/23). We combined open data from two previously published experiments (Meder et al.[30] ($n = 52$, $M_{age} = 6.35$, s.d. = 0.95, 25 female) and Schulz et al.[27] ($n = 79$, $M_{age} = 16.67$, s.d. = 12.82, 33 female) targeting children and adult participants, together with new unpublished data targeting adolescent participants ($n = 150$, $M_{age} = 16.1$, s.d. = 4.97, 69 female). The experimental designs differed in a few details, the majority of which were removed by filtering participants. The combined and filtered data consisted of 281 participants between the ages of 5 and 55 ($M_{age} = 14.46$, s.d. = 8.61, 126 female). Informed consent was obtained from all participants or their legal guardians before participation.

**Generic materials and procedure.** All participants performed a spatially correlated bandit task[38] on an $8 \times 8$ grid of 64 options (that is, tiles). A random tile was revealed at the beginning of each round, with

participants given a limited search horizon of 25 trials to acquire as many cumulative rewards as possible by choosing either new or previously revealed tiles. After each round, participants were rewarded a maximum of five stars reflecting their performance relative to always selecting the optimal tile. The number of stars earned in each round stayed visible until the end of the experiment.

When choosing a tile, participants earned rewards corrupted by normally distributed noise $\epsilon \sim \mathcal{N}(0, 1)$. Reward expectations were spatially correlated across the grid, such that nearby tiles had similar reward expectations (described below). Earned rewards were depicted numerically along with a corresponding colour (colours only in Meder et al.[30]; see below), with darker colours depicting higher rewards. Figure 1a provides a screenshot of the task and Fig. 1b depicts the distribution of rewards on a fully revealed environment.

All experiments (after filtering, see next section) used the same set of 40 underlying reward environments, which define a bivariate function on the grid, mapping each tile's location on the grid to an expected reward value. The environments were generated by sampling from a multivariate Gaussian distribution $\sim \mathcal{N}(0, \Sigma)$, with covariance matrix $\Sigma$ defined by a radial basis function kernel (equation (2)) with $\lambda = 4$. In each round, a new environment was chosen without replacement from the list of environments. To prevent participants from knowing when they found the highest reward, a different maximum range was sampled from a uniform distribution $\sim \mathcal{U}(30, 40)$ for each round and all reward values were rescaled accordingly. The rescaled rewards were then shifted by +5 to avoid reward observations below 0. Hence, the effective rewards ranged from 5 to 45, with a different maximum in each round. All experiments included an initial training round designed to interactively explain the nature of the task, and ended with a bonus round in which they were asked to predict the rewards of unseen tiles. All analyses exclude the training and bonus rounds.

**Differences across experiments.** Participants from the Meder et al.[30] and Schulz et al.[64] studies were recruited from museums in Berlin and paid with stickers (Meder et al.[30]) or with money (Schulz et al.[64]) proportional to their performance in the task. The new adolescent data was collected at the Max Planck Institute for Human Development in Berlin along with a battery of ten other decision-making tasks on a desktop computer. These participants were given a fixed payment of €10 per hour.

The studies by both Meder et al.[30] and Schulz et al.[64] used a between-subject manipulation of the strength of rewards correlations (smooth versus rough environments: $\lambda_{smooth} = 4$, $\lambda_{rough} = 1$). Because only minimal differences in model parameters were found in previous studies[27,30,38], the rough condition was omitted in the adolescent sample. Thus, we filtered out all participants assigned to the rough condition such that only participants assigned to the smooth environments were included in the final sample. Lastly, both Schulz et al.[64] and the adolescent experiment used ten rounds, while Meder et al.[30] included only six rounds to avoid lapses in attention in the younger age group. In addition, numerical depictions of rewards were removed in the Meder et al.[30] experiment, and participants were instructed to focus on the colours (deeper red indicating more rewards) to avoid difficulties with reading large numbers.

After filtering, the remaining differences in modality (tablet versus computer), incentives (stickers versus variable money versus fixed money), number of rounds (six versus ten) and visualization of rewards (numbers plus colours versus colours only) did not result in any differences in performance (Supplementary Fig. 1a), model fits (Supplementary Fig. 1b) or parameter estimates (Supplementary Fig. 1c).

## Computational models

### Gaussian process generalization.
GP regression[40] provides a non-parametric Bayesian framework for function learning, which we use as a method of value generalization[38]. We use the GP to infer a value

functions $f : \mathcal{X} \to \mathbb{R}^n$ mapping input space $\mathcal{X}$ (all possible options on the grid) to a real-valued scalar outputs $r$ (reward expectations). The GP performs this inference in a Bayesian manner, by first defining a prior distribution over functions $p(r_0)$, which is assumed to be multivariate Gaussian:

$$p(r_0(\mathbf{x})) \sim \mathcal{GP}(m(\mathbf{x}), k(\mathbf{x}, \mathbf{x}')), \quad (5)$$

with the prior mean $m(\mathbf{x})$ defining the expected output of input $\mathbf{x}$, and with covariance defined by the kernel function $k(\mathbf{x}, \mathbf{x}')$, for which we use a radial basis function kernel (equation (2)). Per convention, we set the prior mean to zero, without loss of generality[40].

Conditioned on a set of observations $\mathcal{D}_t = \{\mathbf{X}_t, \mathbf{y}_t\}$, the GP computes a posterior distribution $p(r_t(\mathbf{x}_*) | \mathcal{D}_t)$ (equation (1)) for some new input $\mathbf{X}_*$, which is also Gaussian, with posterior mean and variance defined as:

$$m(\mathbf{x}_* | \mathcal{D}_t) = \mathbf{k}_{*,t}^{\top} (\mathbf{K} + \sigma_{\epsilon}^2 \mathbf{I})^{-1} \mathbf{y}_t, \quad (6)$$

$$v(\mathbf{x}_* | \mathcal{D}_t) = k(\mathbf{x}_*, \mathbf{x}_*) - \mathbf{k}_{*,t}^{\top} (\mathbf{K} + \sigma_{\epsilon}^2 \mathbf{I})^{-1} \mathbf{k}_{*,t}, \quad (7)$$

where $\mathbf{K}_{*,t} = k(\mathbf{X}_t, \mathbf{X}_*)$ is the covariance matrix between each observed input and the new input $\mathbf{X}_*$ and $\mathbf{K} = k(\mathbf{X}_t, \mathbf{X}_t)$ is the covariance matrix between each pair of observed inputs. $\mathbf{I}$ is the identity matrix and $\sigma_{\epsilon}^2$ is the observation variance, corresponding to assumed independent and identically distributed Gaussian noise on each reward observation.

**Lesioned models.** The $\lambda$ lesion model removes the capacity for generalization, by replacing the GP with a Bayesian mean tracker (BMT) as a reinforcement learning model that learns reward estimates for each option independently using the dynamics of a Kalman filter with time-invariant rewards. Reward estimates are updated as a function of prediction error, and thus the BMT can be interpreted as a Bayesian variant of the classic Rescorla–Wagner model[76,77] and has been used to describe human behaviour in a variety of learning and decision-making tasks[54,78,79].

The BMT also defines a Gaussian prior distribution of the reward expectations, but does so independently for each option $\mathbf{x}$:

$$p(r_0(\mathbf{x})) \sim \mathcal{N}(m_0(\mathbf{x}), v_0(\mathbf{x})). \quad (8)$$

The BMT computes an equivalent posterior distribution for the expected reward for each option (equation (1)), also in the form of a Gaussian, but where the posterior mean $m_t(\mathbf{x})$ and posterior variance $v_t(\mathbf{x})$ are defined independently for each option and computed by the following updates:

$$m_{t+1}(\mathbf{x}) = m_t(\mathbf{x}) + \delta_t(\mathbf{x}) G_t(\mathbf{x}) (y_t(\mathbf{x}) - m_t(\mathbf{x})), \quad (9)$$

$$v_{t+1}(\mathbf{x}) = v_t(\mathbf{x}) (1 - \delta_t(\mathbf{x}) G_t(\mathbf{x})). \quad (10)$$

Both updates use $\delta_t(\mathbf{x}) = 1$ if option $\mathbf{x}$ was chosen on trial $t$, and $\delta_t(\mathbf{x}) = 0$ otherwise. Thus, the posterior mean and variance are only updated for the chosen option. The update of the mean is based on the prediction error $y_t(\mathbf{x}) - m_t(\mathbf{x})$ between observed and anticipated reward, while the magnitude of the update is based on the Kalman gain $G_t(\mathbf{x})$:

$$G_t(\mathbf{x}) = \frac{v_t(\mathbf{x})}{v_t(\mathbf{x}) + \theta_{\epsilon}^2}, \quad (11)$$

analogous to the learning rate of the Rescorla–Wagner model. Here, the Kalman gain is dynamically defined as a ratio of variance terms, where $v_t(\mathbf{x})$ is the posterior variance estimate and $\theta_{\epsilon}^2$ is the error

variance, which (analogous to the GP) models the level of noise associated with reward observations. Smaller values of $\theta_\epsilon^2$ thus result in larger updates of the mean.

The $\beta$ lesion simply fixes $\beta = 0$, making the valuation of options solely defined by the expected rewards $q(\mathbf{x}) = m(\mathbf{x})$.

The $\tau$ lesion model swaps the softmax policy (characterized by decision temperature $\tau$) for an epsilon-greedy policy[39], since it is not feasible to simply remove the softmax component or fix $\tau = 0$ making it an argmax policy (due to infinite log loss from zero probability predictions). Instead, we take the opportunity to compare the softmax policy against epsilon-greedy as an alternative mechanism of random exploration[28]. We still combine epsilon-greedy with GP and UCB components, but rather than choosing options proportional to their UCB value, the $\tau$ lesion estimates $\epsilon$ as a parameter controlling the probability of choosing an option at random versus the highest UCB option:

$$p(\mathbf{x}) = \begin{cases} \arg\max q(\mathbf{x}), & \text{with probability } 1 - \epsilon \\ 1/64, & \text{with probability } \epsilon \end{cases}. \quad (12)$$

**Model cross-validation.** Each model was fitted using leave-one-round-out cross-validation for each individual participant using maximum likelihood estimation. Model fits are described using negative log likelihoods summed over all out-of-sample predictions, while individual participant parameter estimates are based on averaging over the cross-validated maximum likelihood estimations. Figure 3c reports model fits in terms of a pseudo-$R^2$, which compares the out-of-sample negative log likelihoods for each model $k$ against a random model:

$$R^2 = 1 - \frac{\log \mathcal{L}(M_k)}{\log \mathcal{L}(M_{\text{random}})}. \quad (13)$$

#### Changepoint regression
We use a hierarchical Bayesian changepoint model to quantify univariate changes in model parameters as a function of age:

$$\text{estimate} \sim \mathcal{N}(\mu_{\text{age}}, \sigma^2), \quad (14)$$

$$\mu_{\text{age}} = b_0 + b_1 \times (\text{age} - \omega) \times I(\text{age} \le \omega) \\ + b_2 \times (\text{age} - \omega) \times I(\text{age} > \omega). \quad (15)$$

$I(\cdot)$ is an indicator function, $b_0$ is the intercept, and $\omega$ is the age at which the slope $b_1$ changes to $b_2$. We included random intercepts for different experiments. To account for potential outliers in the parameter estimates, we used a student-$t$ likelihood function as a form of robust regression. Supplementary Table 5 depicts a model comparison between this robust regression reported in the main text, a regression with a Gaussian likelihood that attenuates skew by log transforming the dependent variable and regressions with a Gaussian likelihood that does not account for skew in the dependent variable. Using approximate leave-one-out cross-validation we found that the robust regression consistently fitted the parameter distributions best.

#### Fitness landscape
We used Tukey's fence to define a credible interval for each GP-UCB parameter ($\lambda, \beta, \tau$) based on participant estimates and created a grid of 100 equally sized log-space intervals for each parameter. This defines a space of plausible learning strategies corresponding to one million parameter combinations. We then ran 100 simulations of the GP-UCB model for each parameter combination (sampling one of the 40 reward environments with replacement each time) and computed the mean reward across iterations (Supplementary Fig. 7).

#### Optimization algorithms
Using this fitness landscape defined across learning strategies, we simulated the trajectories of various optimization algorithms. Specifically, we tested SA[4,9] and SHC[12,13], the latter of which provides a discrete analogue to the better-known stochastic gradient descent method commonly used to optimize neural networks[10,11]. Each optimization algorithm (SA versus SHC) was combined with one of three common cooling schedules[80] defining how the optimization temperature (temp) changes as a function of the iteration number $i$. Fast cooling uses $\text{temp}_i = 1/(1 + i)$, exponential cooling uses $\text{temp}_i = \exp(-i^{1/3})$ and linear cooling uses $\text{temp}_i = 1 - (i + 1)/\max(i)$. As the optimization temperature decreases over iterations, there is a general decrease in the amount of randomness or stochasticity.

SA is a stochastic sampling algorithm, which is more likely to select solutions with lower fitness when the optimization temperature is high. After initialization, SA iteratively selects a random neighbouring solution $s_{\text{new}}$ in the fitness landscape (that is, one step in the grid of one million parameter combinations), and deterministically accepts it if it corresponds to higher fitness than the current solution $s_{\text{old}}$; otherwise, it accepts worse solutions with probability:

$$p(\text{accept}) \propto \exp\left(\frac{s_{\text{new}} - s_{\text{old}}}{\text{temp}_i}\right), \quad (16)$$

where $\text{temp}_i$ is the current optimization temperature.

SHC is similar, but considers all neighbouring solutions $s' \in \mathcal{S}_{\text{neighbors}}$ and selects a new solution proportional to its fitness:

$$p(s') \propto \exp(s'/\text{temp}_i). \quad (17)$$

For each combination of optimization algorithm and cooling function, we simulated optimization trajectories over 1500 iterations. Each simulated was initialized on each of the cross-validated parameter estimates of all participants of the youngest age group as starting points. This resulted in 120 (30 participants × 4 rounds of cross-validation) trajectories for each combination of algorithm and cooling schedule.

#### Reporting summary
Further information on research design is available in the Nature Portfolio Reporting Summary linked to this article.

## Data availability
Data are publicly available at https://github.com/AnnaGiron/developmental_trajectory.

## Code availability
Code is publicly available at https://github.com/AnnaGiron/developmental_trajectory.

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

## Acknowledgements

We thank K. Murayama, M. Sakaki, P. Schwartenbeck, M. Hamidi and R. Uchiyama for helpful feedback and C. Wysocki for data collection. This work was supported by the German Federal Ministry of Education and Research (BMBF): Tübingen AI Center, FKZ: 01IS18039A (C.M.W.) and funded by the Deutsche Forschungsgemeinschaft (DFG, German Research Foundation) under Germany's Excellence Strategy-EXC2064/1-390727645 (C.M.W.).

## Author contributions

C.M.W., S.C. and A.P.G. conceived the study with feedback from all authors. S.C. collected the data using materials provided by C.M.W. C.M.W., A.P.G. and S.C. performed the analyses. C.M.W., S.C. and A.P.G. wrote the paper, with feedback from E.S., W.B., A.R. and B.M.

## Funding

## Competing interests

The authors declare no competing interests.

## Additional information

**Correspondence and requests for materials** should be addressed to Charley M. Wu.

# Reporting Summary

## Statistics

For all statistical analyses, confirm that the following items are present in the figure legend, table legend, main text, or Methods section.

| n/a | Confirmed | |
|---|---|---|
| ☐ | ☒ | The exact sample size (*n*) for each experimental group/condition, given as a discrete number and unit of measurement |
| ☐ | ☒ | A statement on whether measurements were taken from distinct samples or whether the same sample was measured repeatedly |
| ☐ | ☒ | The statistical test(s) used AND whether they are one- or two-sided<br>*Only common tests should be described solely by name; describe more complex techniques in the Methods section.* |
| ☐ | ☒ | A description of all covariates tested |
| ☐ | ☒ | A description of any assumptions or corrections, such as tests of normality and adjustment for multiple comparisons |
| ☐ | ☒ | A full description of the statistical parameters including central tendency (e.g. means) or other basic estimates (e.g. regression coefficient) AND variation (e.g. standard deviation) or associated estimates of uncertainty (e.g. confidence intervals) |
| ☐ | ☒ | For null hypothesis testing, the test statistic (e.g. $F$, $t$, $r$) with confidence intervals, effect sizes, degrees of freedom and $P$ value noted<br>*Give P values as exact values whenever suitable.* |
| ☐ | ☒ | For Bayesian analysis, information on the choice of priors and Markov chain Monte Carlo settings |
| ☐ | ☒ | For hierarchical and complex designs, identification of the appropriate level for tests and full reporting of outcomes |
| ☐ | ☒ | Estimates of effect sizes (e.g. Cohen's *d*, Pearson's *r*), indicating how they were calculated |

*Our web collection on statistics for biologists contains articles on many of the points above.*

## Software and code

Policy information about availability of computer code

| | |
|---|---|
| Data collection | We used online experiments, where experiment code is freely available at https://github.com/charleywu/gridsearch and https://github.com/ericschulz/kwg. Cordova version xy was used to implement the experiment. |
| Data analysis | All code used to analyze the data is freely available at https://github.com/AnnaGiron/developmental_trajectory. We used R version 4.0.3 and Python 3.8 for the analyses. |

For manuscripts utilizing custom algorithms or software that are central to the research but not yet described in published literature, software must be made available to editors and reviewers. We strongly encourage code deposition in a community repository (e.g. GitHub). See the Nature Portfolio guidelines for submitting code & software for further information.

## Data

Policy information about availability of data

All manuscripts must include a data availability statement. This statement should provide the following information, where applicable:
- Accession codes, unique identifiers, or web links for publicly available datasets
- A description of any restrictions on data availability
- For clinical datasets or third party data, please ensure that the statement adheres to our policy

Experiment data is freely available at https://github.com/AnnaGiron/developmental_trajectory.

# Research involving human participants, their data, or biological material

Policy information about studies with human participants or human data. See also policy information about sex, gender (identity/presentation), and sexual orientation and race, ethnicity and racism.

| | |
|---|---|
| Reporting on sex and gender | We did not perform sex- or gender-based analysis because we did not expect any sex- or gender-based differences. |
| Reporting on race, ethnicity, or other socially relevant groupings | We did not collect any data on race, ethnicity or other socially relevant groupings. |
| Population characteristics | See below. |
| Recruitment | Participants from Meder et al. (2021) and Schulz et al. (2019) were recruited in museums in Berlin. The new adolescent participants performed the task in the Max Planck Institute for Human Development in Berlin. |
| Ethics oversight | All studies were approved by the ethics board of the Max Planck Institute for Human Development in Berlin. |

Note that full information on the approval of the study protocol must also be provided in the manuscript.

# Field-specific reporting

Please select the one below that is the best fit for your research. If you are not sure, read the appropriate sections before making your selection.

☐ Life sciences  ☒ Behavioural & social sciences  ☐ Ecological, evolutionary & environmental sciences

For a reference copy of the document with all sections, see [nature.com/documents/nr-reporting-summary-flat.pdf](http://nature.com/documents/nr-reporting-summary-flat.pdf)

# Behavioural & social sciences study design

All studies must disclose on these points even when the disclosure is negative.

| | |
|---|---|
| Study description | The data presented here are a combination of three different datasets (Schulz et al., 2019, Meder et al., 2021 and an unpublished dataset). In all experiments, we collected quantitative data based on individual's choice in each trial. Participants were first given instructions for the task along with several examples of fully revealed environments. Then they were asked to complete a set of comprehension questions before starting the experiment. |
| Research sample | The data were collected in three different experiments, two of which have been published before (Meder el at., 2021 and Schulz et al., 2019) and one unpublished dataset targeting adolescent participants. Participants from the unpublished dataset completed the task at the Max Planck Institute for Human Development in Berlin along with a battery of 10 other decision-making tasks on a desktop computer. These participants were given a fixed payment of 10€ per hour. The sample size consists of 281 participants between the ages of 5 and 55 (mean age=14.46, sd=8.61, 126 female). |
| Sampling strategy | The sample size was chosen to be comparable to the previously published datasets (Schulz et al., 2019, Meder et al., 2021) and appropriately scaled for the larger age range. Since the focus of our analysis was on computational modeling, rather than purely behavioral analysis, our sample size determination was not focused on achieving the necessary power to observe a specific effect size. |
| Data collection | Data from Schulz et al. (2019) and Meder et al (2021) were collected in museums in Berlin and participants performed the task on a tablet. Participants from the unpublished dataset were recruited via phone interview from the internal database of the Max Planck Institute for Human Development. The experimenter explained the task and then left the room while participants performed the experiment along with a battery of 10 other decision-making experiments on a desktop computer. For all datasets, the experimenter was blinded regarding to the study hypothesis. |
| Timing | The new data was collected between October 2018 and June 2019. |
| Data exclusions | In the original Meder et al., dataset, 14 participants were excluded due to failing the instruction check (n=9), did not complete the task (n=1), were not native speakers (n=2), or because their parents intervened during the experiment (n=2). In the Schulz et al. and the unpublished dataset, no collected data were excluded. For the joint analysis of the different datasets, we filtered the dataset from Meder et al. and Schulz et al. to only use participants assigned to one of the experimental conditions (smooth environments). In doing so, we removed 50 participants from Meder et al. and 81 participants from Schulz et al. |
| Non-participation | We had no non-participants. |
| Randomization | Participants were not allocated into experimental groups. |

# Reporting for specific materials, systems and methods

We require information from authors about some types of materials, experimental systems and methods used in many studies. Here, indicate whether each material, system or method listed is relevant to your study. If you are not sure if a list item applies to your research, read the appropriate section before selecting a response.

## Materials & experimental systems

| n/a | Involved in the study |
|-----|-----------------------|
| ☒ ☐ | Antibodies |
| ☒ ☐ | Eukaryotic cell lines |
| ☒ ☐ | Palaeontology and archaeology |
| ☒ ☐ | Animals and other organisms |
| ☒ ☐ | Clinical data |
| ☒ ☐ | Dual use research of concern |
| ☒ ☐ | Plants |

## Methods

| n/a | Involved in the study |
|-----|-----------------------|
| ☒ ☐ | ChIP-seq |
| ☒ ☐ | Flow cytometry |
| ☒ ☐ | MRI-based neuroimaging |

