## [Peer Review File · Nature Human Behaviour]

Peer Review Information

Journal: Nature Human Behaviour

Manuscript Title: Developmental changes in exploration resemble stochastic optimization

Corresponding author name(s): Charley M. Wu

Reviewer Comments & Decisions:

Decision Letter, initial version:

13th January 2023

Dear Dr Wu,

Thank you once again for your manuscript, entitled "Developmental changes resemble stochastic optimization," and for your patience during the peer review process.

Your manuscript has now been evaluated by 3 reviewers, whose comments are included at the end of this letter. Although the reviewers find your work to be of high interest, they also raise some important concerns. We are interested in the possibility of publishing your study in Nature Human Behaviour, but would like to consider your response to these concerns in the form of a revised manuscript before we make a decision on publication.

In particular, you will see that both Reviewer #2 and #3 raise the question of whether the developmental results might be explained by changes in aspects of performance other than exploration, such as lack of motivation in young children. It will be important to address this with further analysis.

In sum, we invite you to revise your manuscript taking into account all reviewer and editor comments. We are committed to providing a fair and constructive peer-review process. Do not hesitate to contact us if there are specific requests from the reviewers that you believe are technically impossible or unlikely to yield a meaningful outcome.

We hope to receive your revised manuscript within two months. I would be grateful if you could contact us as soon as possible if you foresee difficulties with meeting this target resubmission date.

- Include a "Response to the editors and reviewers" document detailing, point-by-point, how you addressed each editor and referee comment. If no action was taken to address a point, you must provide a compelling argument. When formatting this document, please respond to each reviewer comment individually, including the full text of the reviewer comment verbatim followed by your

response to the individual point. This response will be used by the editors to evaluate your revision and sent back to the reviewers along with the revised manuscript.

- Highlight all changes made to your manuscript or provide us with a version that tracks changes.

[REDACTED]

We look forward to seeing the revised manuscript and thank you for the opportunity to review your work. Please do not hesitate to contact me if you have any questions or would like to discuss these revisions further.

Sincerely,
Jamie

Dr Jamie Horder
Senior Editor
Nature Human Behaviour

REVIEWER COMMENTS:

Reviewer #1:
Remarks to the Author:

This is a very good paper. It extends a compelling and influential series of articles studying human exploration and generalization through the spatially correlated bandits task. Rather than studying just adults (or children) as in previous work, this paper aims to characterize changes across the lifespan. It compares how hyperparameters change over time (using fits from sophisticated model-based behavioral analyses) compared to the same parameters chosen via standard stochastic optimization algorithms. The research is well-designed, thoroughly analyzed, and clearly presented. The technical work and modeling is very strong. The comparison with stochastic optimization is well-done and thought provoking, adding quantitative heft compared to the more informal comparisons that are typical in the literature.

The paper has many strengths, but there are ways it could be improved:

The comparison between human development and stochastic optimization is central to the paper, but the comparison is largely done with mean values only. The figures illustrating the trajectories (Fig 4B and S8) show only median parameter values for the algorithms and mean values for the people. Given that the GP-UCB model is fit to individual participants (which is great!), and that different runs of the stochastic optimization can lead to different trajectories, to understand the depth of the analogy it seems essential to see how the variability compares as well. Fig. S8 shows the human variability, which is helpful, but not the algorithm variability (the optimizers are stochastic, after all!). Crucially, one could ask: *How often do runs of the algorithm reach the region of adult parameter values?*

Is it right that simulated annealing (and hill climbing) algorithms are searching across temperature parameters in the GP-UCB model (along with generalization and exploration parameters), while these algorithms themselves *also* have a temperature hyperparameter which is adjusted on a schedule? It was non-trivial to wrap my head around this, so this distinction could be made more explicit and clear.

Equally, for previous (perhaps simplistic, as you argue) suggestions that development is like a decreasing temperature--which temperature would be the appropriate one to compare? Indeed, the higher-level stochastic optimization temperature does decrease over time on a simple schedule. In short, I would appreciate more discussion on the psychological meaning of the different parameters, especially the two different temperatures.

In what ways would you expect your findings regarding the parameters for generalization, exploration, and temperature to generalize to other tasks? Here, the hypothesis space is a smooth grid of numbers, and learning occurs over a sequence of trials in an experiment. If you were to speculate, would you find similar results for search over structured hypotheses, say over different intuitive theories or different programs (using the child as hacker analogy you cite in the conclusion)? What about for a search over hypotheses / intuitive theories that may take years rather than minutes? I would love to see some discussion of these possibilities.

A couple more minor things:

- In the last paragraph of the introduction, it's hard to follow for each sentence whether you are talking about A) stochastic search for the best hypothesis given a fix posterior distribution $P(H|D)$ with fixed D , or B) how the posterior $P(H|D)$ changes and tightens with more data D over time (regardless of inference/search algorithm). One is about stochastic optimization algorithms and the other is not.

- Your discussion of previous work on temperature, generalization, and exploration in development is well done. It would be especially helpful to specifically highlight which results were obtained using the same spatially correlated bandits task as used here, and thus more clearly specify how this paper builds on this previous work. I'm also a little confused regarding which datasets were previous datasets were combined for analysis in this paper. The main text cites ref 27 and 28 while the Methods cite refs 28 and 69 instead.

Final thoughts:

In sum, this is high quality research that makes a substantial contribution. The article does build on a line of existing research with this task (including papers studying developmental change), and for this reason I wondered whether this paper would more appropriate for a more specialized journal. I could see the case either way, but ultimately I believe it warrants publication in Nature Human Behavior (after appropriate revisions). The authors use an established paradigm, sure, but to great effect. The paper is a strong step forward in its stated goal of bringing much-needed clarity to commonly used analogies of optimization and cognitive development.

Reviewer #2:

Remarks to the Author:

This paper looks at the developmental changes in explore-exploit parameters capturing reward generalization, exploratory bias, and random exploration. The estimation of these parameters and empirical approach is grounded in past research that has primarily focused on preschool and younger elementary children, and occasionally comparing to small group of adults, finding that younger children tend to be more "random" and exploratorily biased than adults or optimal models. Importantly, the work here has extended this prior work to include large sample of middle adolescence and fleshing out adult data. With larger sample and continuous age ranges, the authors are able to take more detailed dive into developmental/age changes over the lifespan. Most importantly, the authors investigate how these parameterizations (the learning strategies) change over time and different within age groups. This analysis thus represents a significant, new contribution to a growing research program seeking to understand the mechanisms that underlie human intelligence.

Overall, the paper is excellently written. The extremely large and age-diverse data sample are

impressive, and the modeling and analyses are sound. The design and approach is appropriate, and I believe the paper will appeal to a large community of this journal's readership. While I have provided a somewhat longwinded review below, I want to be clear that these comments are extremely minor – and mostly amount to a few minor points of clarification and extension that will increase readability for a larger audience.

1. In the introduction, the authors nicely clarify that randomness is different from overall exploratory bias (which is a different parameter). This point is sometimes confused in past literature but is indeed distinct as noted/modeled here. It would be helpful in the section on stochasticity to clarify that randomness can apply to (at least) three different metrics: (a) noise in whether to explore/exploit, (2) noise in where to exploit (if exploiting), and (3) noise in where to explore (if exploring). The randomness parameter τ in this paper captures random exploration (choice stochasticity in exploration location), which is appropriate for this task and similar to past studies from some of the authors of this group modeling this kind of task. This is just one minor clarification suggested for the introduction/overview; it was clear how this conceptual argument aligned with the analyses performed.

2. Due to space limitations, I initially had trouble understanding how the point about changes in generalization and integration of new experiences differed conceptually from the following paragraph on Bayesian narrowing of priors. Providing a bit more space so the authors can clarify and expand on these points would go a long way – as these are complicated and abstract concepts that may be difficult for a less expert audience to grasp in the short-report space allotted. Specifically, it was initially unclear how both of these points aligned with the reported behavioral results, and only became more clear in the model-based analysis section. Generalization is defined here in terms of how observed reward values generalize to nearby unexplored spaces (e.g. the degree of similarity over distance assumed for reward distributions), but (as far as I can tell) is not also a parameter for learning this overall distance distribution over rounds. A brief point of clarification in the introduction re: type of generalization would go a long way. The Bayesian learning strategies are explored in terms of the stochasticity and rate of change of learning parameters (a kind of meta-level analysis). A few additional sentences of clarification and signposting in the introduction would also go a long way for a less expert reader.

3. I also felt more discussion (and possibly additional analysis) could be given to two possible interpretations of this data. The first is that overall for ALL trials/decision making, children are following the explore-exploit decision expectations of the study and the children tend to show overall lower mean and max reward, higher unique options generated, a flattened search distance to reward, and decreased repeating (and increased "near clicks"). A second possibility is that younger children grow tired of the task within round (as they feel they have learned enough about the space), and meta-goal shift to "play" with the squares, no longer attending to the explore-exploit optimization (potentially around trial 15-20 by examination of 2b). It's possible that the explore-exploit behavior of children is quite similar to adults (though still arguably slower in their learning curves) up until this "pooping-out/play-goal-shifting point", but that the decision to no longer engage in the original goals of the task leads to a big shift in behavior later in the task that thus dilutes overall mean and max reward, leads to higher unique options generated, flattened search distance, and decreased repeating (especially if the children are drawing pictures on the board or just clicking for fun). This idea is also related to the "more diverse strategies" point brought into the modeling section. It's possible that overall children's explore-exploit strategies (parameters) differed more from each other than adults overall, but it's also possible that this result falls out because children had more varied interpretations of the overall meta-level task structure (and/or more goal shifting to meta "play" goals, etc within rounds) than adults, which led to this variability. To be clear, either explanation supports the broader points the authors are making in this manuscript and so should not be counted as a potential confound in interpretation. Nonetheless, it would be interesting to see further trial-by-trial analysis to investigate whether younger children hit a "poop-out" inflection point part way in (e.g. maybe splitting analysis for children around trial ~17), to explore whether there is a behavioral shift that might partly explain the overall developmental results. My money is on the bet that the authors will find that overall (even in the first 16 trials) the results/parameters replicate, but that even stronger "random" behavior is observed in the later trials – suggesting that both overall children are more random and exploratory

than adults, but also that they are less likely to stick with a task and “shift” at a goal/meta level part-way through each round that leads to even greater random and exploratory-bias. Nonetheless, I think it would be interesting to attempt to explore whether younger children are following task expectations (making explore-exploit decisions) throughout or whether they are switching up the task expectations all together and just playing or randomly clicking part way through because they lost interest in/value for the reward. This is a challenging analysis proposal because it is exploratory and post-hoc after examining the results – so please take this suggestion with a grain of salt. (Perhaps they make the most sense in a supplementary section.) Nonetheless, I think it is interesting to understand whether there are hints that children are strategy shifting within the task, or whether the results suggest consistent explore-exploit parameters of generalization, exploration bias, and randomness across all the trials.

4. Please remind the readers what tau, lambda, and beta parameters capture whenever possible to help less technical readers. For example, “ λ and β parameters converged at lower values than the algorithm (...), while τ was not credibly higher or lower...” Could include light signposting to help with readability, e.g. “ λ (“generalization”) and β (“exploration bias”) parameters converged at lower values than the algorithm (...), while τ (“exploratory randomness”) was not credibly higher or lower...”

5. A stronger acknowledgement to the relative “WEIRD”ness of the sample, with implications for whether/how these findings could replicate with more representative world samples is needed. Both clarifying the sample characteristics and couching claims in terms of this narrowness are needed. I say this with the firm belief that the development of explore-exploit strategies examined here are likely to generalize across many populations, but also with the understanding that children who have experienced early childhood adversity may show differences in overall explore-exploit thresholds, with a preference for exploitation (Humphreys et al., 2015; Lloyd, McKay, & Furl, 2022). A nod to this work (and need for future work to explore and extend to Socio-economically, racially, globally, and early-stress-experience diverse populations is needed.

I commend the authors on an excellent manuscript and hope they find these minor suggestions helpful in future revisions.

Reviewer #3:
Remarks to the Author:

Review of Developmental changes resemble stochastic optimization by Giron et al

The authors report analysis of an aggregated behavioural dataset in which children, adolescents and adults perform an RL task that requires spatial generalisation. Their question is how exploration (characterised here as a tendency to switch away from previously successful action) varies across the developmental trajectory. They report a number of interesting findings, and in particular (using a modelling approach) that exploration declines and then plateaus in the early teenage years, and that parameter fits become more similar across individuals with age. The former applies both to exploration modelled as a temperature parameter (general randomness) and the uncertainty-weight the computation of value (in a UCB model). The authors interpret their data as consistent with theories that have emphasised “cooling off” during development, with policies become increasingly crystallised over the lifespan.

I thought this was an impressive paper. I learned from it and would cite it. The work is clearly explained and nicely presented. The modelling is accomplished. I was initially worried about whether the authors model was identifiable, but the model / parameter recovery in the supplementary is convincing. It is nice to see lesion studies to show that all three parameters are needed.

The major thought I had as reading the paper is whether the authors’ modelling results would be different if there were a simple improvement in performance across the lifespan due to generic factors

that have nothing to do with exploration (for example, perhaps kids that are > 10 years simply understand the instructions better, or have a longer attention span, or better motor coordination). In particular given that tau and beta follow a similar trajectory it seems possible (although perhaps unlikely due to recovery success) that they are both tapping into a generic decrease in randomness at around 10 years. Here are some suggestions for how the authors could perhaps prevent similar worries in other readers.

1) could the authors perhaps compare models that involve restrictions on the change point separately for the three parameters? If I've understood correctly, here the authors ask whether a model in which there is a single change point for all three parameters fares better than models with no or two or three change points, and presumably flat (intercept only) models. However, could the data be better fit by a model in which only tau is changing (either linearly or with a single change point) but beta and lambda are not? This "restriction analysis" is similar to the lesion analysis but does not ask whether each parameter is required at all, but asks whether each parameter is needed to account for developmental change. This would potentially allow the authors to rule out another model in which there is a single generic factor that changes over development, rather than to the more nuanced model that the authors favour.

2) another route would be to say more about the relationship between exploration and performance. Simulated annealing has a normative explanation, as it's better to explore early and exploit later. But in this study, a tendency to exploit a known source of reward seems to be the reason why adults perform better, which makes the reader feel that perhaps younger participants are just generically doing worse, rather than strategically exploring more. Could the authors show – perhaps capitalising on individual differences but keeping age as a covariate – that there are *benefits* of exploration and that these benefits independently vary with age? For example, how does search distance early in play predict reward harvesting later, and how does this vary between individuals?

3) It's striking to me that the youngest participants are more likely than adults to use a "next to" strategy (search distance = 1) but not a random spatial sampling strategy. Does the mode recreate this finding? If not, what might be needed for it to do so? [minor: what is the dotted red line in Fig 2e? it's not referred to in the legend]. Related to point (2) above, are there benefits of this strategy? Do these benefits vary with age?

So in short I think this is a great paper but I would love it if the authors went a bit further in showing that young people's behaviour is "exploration" and not just "error".

Author Rebuttal to Initial comments

1 Reviewer 1

1.1 General remarks

This is a very good paper. It extends a compelling and influential series of articles studying human exploration and generalization through the spatially correlated bandits task. Rather than studying just adults (or children) as in previous work, this paper aims to characterize changes across the lifespan. It compares how hyperparameters change over time (using fits from sophisticated model-based behavioral analyses) compared to the same parameters chosen via standard stochastic optimization algorithms. The research is well-designed, thoroughly analyzed, and clearly presented. The technical work and modeling is very strong. The comparison with stochastic optimization is well-done and thought provoking, adding quantitative heft compared to the more informal comparisons that are typical in the literature.

We thank the reviewer for the positive feedback on our manuscript.

1.2 Variation of stochastic optimization trajectories

The comparison between human development and stochastic optimization is central to the paper, but the comparison is largely done with mean values only. The figures illustrating the trajectories (Fig 4B and S8) show only median parameter values for the algorithms and mean values for the people. Given that the GP-UCB model is fit to individual participants (which is great!), and that different runs of the stochastic optimization can lead to different trajectories, to understand the depth of the analogy it seems essential to see how the variability compares as well. Fig. S8 shows the human variability, which is helpful, but not the algorithm variability (the optimizers are stochastic, after all!). Crucially, one could ask: *How often do runs of the algorithm reach the region of adult parameter values?*

We thank the reviewer for pointing this out. We included a plot that shows the variability of trajectories for each combination of algorithm and cooling schedule (Fig. S9; also reproduced here as Fig. R1 for convenience). Each combination of algorithm and cooling schedule was initialized with all individual cross-validated parameter estimates of the youngest age group once (30 participants \times 4 rounds), which provides us with four trajectories per participant and algorithm. To visualize the variability of algorithm trajectories, we iteratively dropped one of the trajectories for each participant and then use the same aggregation method (mean for each participant and then the median trajectory across participants; 100 iterations). We then plot each of these leave-one-out aggregated trajectories to visualize the variability of the algorithm trajectories.

Additionally we would like to point the reviewer to Fig. S10c, in which we plot the parameter distribution after the changepoint ω . This analysis provides a comparison of post-convergence parameter distributions of the stochastic optimization algorithms against the oldest age-group of human participants (25-55 year olds). We quote the results in the figure caption, which are also summarized in the main text.

Human λ and β parameters converged at lower values than the algorithm (λ : $U = 2740$, $p < .001$, $r_\tau = -.38$, $BF > 100$; β : $U = 10114$, $p < .001$, $r_\tau = -.19$, $BF > 100$), while τ estimates were not reliably higher or lower ($U = 22377$, $p = .188$, $r_\tau = .05$, $BF = .12$).

Figure R1: **Variability of trajectories of generalization λ and uncertainty-directed exploration β .** Plots show the bootstrapped variability of trajectories (blue) as well as the median trajectories (black) for each combination of optimization algorithm (rows) and cooling function (columns).

1.3 Difference between temperature parameters

Is it right that simulated annealing (and hill climbing) algorithms are searching across temperature parameters in the GP-UCB model (along with generalization and exploration parameters), while these algorithms themselves *also* have a temperature hyperparameter which is adjusted on a schedule? It was non-trivial to wrap my head around this, so this distinction could be made more explicit and clear. Equally, for previous (perhaps simplistic, as you argue) suggestions that development is like a decreasing temperature—which temperature would be the appropriate one to compare? Indeed, the higher-level stochastic optimization temperature does decrease over time on a simple schedule. In short, I would appreciate more discussion on the psychological meaning of the different parameters, especially the two different temperatures.

We agree with the reviewers' notion that the temperature parameter of the GP-UCB model and the temperature parameter of the optimization algorithm are at risk of confusion in our manuscript. We are happy that this point is raised in the first place, as it shows that we successfully addressed an important conceptual distinction between different temperature parameters that previously lacked clarity in the literature (Meder, Wu, Schulz, & Ruggeri, 2021; Schulz, Wu, Huys, Krause, & Speekenbrink, 2018). However, we could have been more clear. We now invoke the distinction between *decision temperature* and *optimization temperature* throughout the text. The former is psychologically specific to each individual at a point in time. The latter describes the stochasticity of change of all parameters at the population level, and this is what we believe describes the stochastic optimization analogy best.

We are grateful for this comment because one of the motivations of our paper is to resolve existing ambiguity regarding how to interpret the simulated annealing metaphor, and we believe precise terminology is key.

1.4 Generalization to other tasks

In what ways would you expect your findings regarding the parameters for generalization, exploration, and temperature to generalize to other tasks? Here, the hypothesis space is a smooth grid of numbers, and learning occurs over a sequence of trials in an experiment. If you were to speculate, would you find similar results for search over structured hypotheses, say over different intuitive theories or different programs (using the child as hacker analogy you cite in the conclusion)? What about for a search over hypotheses / intuitive theories that may take years rather than minutes? I would love to see some discussion of these possibilities.

We thank the reviewer for bringing up the issue of generalization to other tasks. The suggestion of speculating about search over intuitive theories or programs is indeed quite intriguing, since we could potentially model these domains as search over some latent hypothesis space. Already, some previous work has used the same GP-UCB model to describe behavior across both graph-structured (Wu, Schulz, & Gershman, 2021) and abstract, conceptual (Wu, Schulz, Garvert, Meder, & Schuck, 2020) domains, finding similar parameter ranges as here.

In Wu et al. (2021), participants searched on graph-structured environments where rewards were defined by network connectivity rather than spatial distance. The same GP-UCB model best predicted choices, but using a graph kernel that extends the RBF kernel to structured domains. Additionally, Wu et al. (2020) used a within-subject task where participants used arrow keys to either select a spatial location (equivalent to the

task here but with a different input mechanism) or to modify the abstract features of a Gabor patch (tilt and stripe frequency). The same GP-UCB model provided the best fit in both tasks. And whereas we found no changes in the generalization parameter λ (also similar ranges to this work), the Gabor task resulted in reduced directed exploration (β) and increased random exploration (τ).

Thus, the framework appears to be robust to changes in the generative reward process (spatial vs. graph-structured) and input modality (mouse clicks, touch screen, and arrow keys), although the presentation of the search space (spatial vs. visual features) appears to influence exploration strategies.

Extending this to search over latent hypothesis spaces should in principle be possible, provided we are able to reliably infer some latent representation about which unobservable hypotheses are being considered, given the observable actions. This suggestion also dovetails with our proposal for needing richer paradigms in order to perform longitudinal analyses, where we can measure these dimensions of learning over sufficiently distinct tasks administered at different developmental stages. Thus, we have added the following text in the discussion:

Yet future longitudinal analyses may be possible using a richer paradigm. For instance, modeling how we learn intuitive theories about the world (Gerstenberg & Tenenbaum, 2017) or compositional programs (Rule, Tenenbaum, & Piantadosi, 2020) as a search process in some latent hypothesis space. The richness of these domains may allow similar dimensions of learning to be measured from sufficiently distinct tasks administered at different developmental stages.

1.5 Bayesian theories of development

In the last paragraph of the introduction, it's hard to follow for each sentence whether you are talking about A) stochastic search for the best hypothesis given a fixed posterior distribution $P(H|D)$ with fixed D , or B) how the posterior $P(H|D)$ changes and tightens with more data D over time (regardless of inference/search algorithm). One is about stochastic optimization algorithms and the other is not.

We apologize for being unclear in that regard and agree that the link to Bayesian development theories cannot be made without further elaboration. We are grateful for the chance to improve our manuscript in response to this and Reviewer 2's similar comment (Sec 3.3).

In the article, we show that the stochastic optimization analogy is best interpreted with respect to changes in learning strategy (quantified by GP-UCB parameters) over the lifespan. The gradual reduction of stochasticity over training iterations is thus analogous to the gradual convergence of participant parameters over development.

Similarly, sequential Bayesian updating also displays similar patterns of gradual convergence. As people acquire more data over their lifespan, Bayesian principles predict a gradual narrowing of the posterior, which should, in turn, lead to less stochasticity in how they deploy hypotheses (e.g., about which strategy to use in a given environment) and select actions. And while we do not have direct access to the life history data of participants before they participated in our experiment, we can illustrate this general principle with a toy problem using MCMC sampling (see figure R2). Each Markov chain initially starts with highly diverse estimates but then converges over time. Thus, Bayesian models for cognitive development offer another potential theoretical lens to interpret our results rather than being mutually exclusive from the stochastic optimization analogy.

We have now modified the text in the introduction to clarify how stochastic optimization and Bayesian theories of development are distinct theoretical frameworks but are linked via similar predictions about gradual convergence:

Alternatively, one could apply the "cooling off" metaphor to an optimization process in the space of learning strategies, which can be characterized across multiple dimensions of learning. Development might thus be framed as parameter optimization, which tunes the parameters of an

individual's learning strategy, starting off by making large tweaks in childhood, followed by gradually lesser and more refined adjustments over the lifespan. In the metaphor, training iterations of the algorithm become a proxy for age.

This interpretation connects the metaphor of stochastic optimization with Bayesian models of cognitive development, which share a common notion of gradual convergence (Stamps & Frankenhuis, 2016; Tenenbaum, Kemp, Griffiths, & Goodman, 2011). In Bayesian models of development, individuals in early developmental stages possess broad priors and vague theories about the causal structure of the world, which become refined with experience (Tenenbaum et al., 2011). Bayesian principles dictate that over time, novel experiences will have a lesser impact on future beliefs or behavior as one's priors become more narrow (Frankenhuis & Panchanathan, 2011; Stamps & Frankenhuis, 2016). Observed over the lifespan, this process will result in large changes to beliefs and behavior early in childhood and smaller changes in later stages, implying a similar developmental pattern as the stochastic optimization metaphor. In sum, not only might the outcomes of behavior be more stochastic during childhood, but the changes to the parameters governing behavior might also be more stochastic in earlier developmental stages.

1.6 Previous datasets

Your discussion of previous work on temperature, generalization, and exploration in development is well done. It would be especially helpful to specifically highlight which results

Figure R2: **Convergence of Bayesian inference.** a) MCMC sampling using the Metropolis-Hastings algorithm, estimating the population parameters of 100 samples from a normal distribution with $\mu = 10$ and $\sigma = 4$. Each line is one Markov chain, where we initially see high variability in the hypotheses about the population parameter μ (y-axis) in early sampling iterations (x-axis), which then gradually converges. b) The posterior summary statistics that we would obtain after stopping the algorithm at 350 iterations

(Early) or discarding the first 350 iterations as burn-in (Late).

were obtained using the same spatially correlated bandits task as used here, and thus more clearly specify how this paper builds on this previous work. I'm also a little confused regarding which datasets were previous datasets were combined for analysis in this paper. The main text cites ref 27 and 28 while the Methods cite refs 28 and 69 instead.

We thank the reviewer for this positive assessment and for pointing out this issue with clarity and for finding the incorrect citation. We have corrected the citation in the methods, which should resolve any confusion about which datasets were combined.

We agree that it would be helpful to clarify which results in the intro were based on our task vs. other tasks. We focused on providing an accurate picture of what the field largely agrees on, which warrants a diverse selection of references. Since our spatially correlated bandit paradigm is not yet introduced at this point, it does not seem appropriate to use the name of the task to single out our previous work. However, we have now amended the text in "Goals and Scope" to distinguish between our work and the work of others in terms of whether the tasks used structured rewards, although we don't go into greater detail about the many new analyses presented here:

While past work has compared differences in parameters between discrete age-groups in both structured (Meder et al., 2021; Schulz, Wu, Ruggeri, & Meder, 2019) and unstructured reward learning tasks (Blanco & Sloutsky, 2021; Somerville et al., 2017; Van den Bos, Cohen, Kahnt, & Crone, 2012), here we characterize the shape of developmental change across the lifespan, from ages 5 to 55.

1.7 Summary

In sum, this is high quality research that makes a substantial contribution. The article does build on a line of existing research with this task (including papers studying developmental change), and for this reason I wondered whether this paper would more appropriate for a more specialized journal. I could see the case either way, but ultimately I believe it warrants publication in *Nature Human Behavior* (after appropriate revisions). The authors use an established paradigm, sure, but to great effect. The paper is a strong step forward in its stated goal of bringing much-needed clarity to commonly used analogies of optimization and cognitive development.

We are incredibly grateful for the positive evaluation of this work and the thoughtful feedback.

2 Reviewer 2

2.1 General remarks

This paper looks at the developmental changes in explore-exploit parameters capturing reward generalization, exploratory bias, and random exploration. The estimation of these parameters and empirical approach is grounded in past research that has primarily focused on preschool and younger elementary children, and occasionally comparing to small group of adults, finding that younger children tend to be more "random" and exploratorily biased than adults or optimal models. Importantly, the work here has extended this prior work to include large sample of middle adolescence and fleshing out adult data. With larger sample and continuous age ranges, the authors are able to take more detailed dive into developmental/age changes over the lifespan. Most importantly, the authors investigate how these parameterizations (the learning strategies) change over time and different within age groups. This analysis thus represents a significant, new contribution

to a growing research program seeking to understand the mechanisms that underlie human intelligence.

Overall, the paper is excellently written. The extremely large and age-diverse data sample are impressive, and the modeling and analyses are sound. The design and approach is appropriate, and I believe the paper will appeal to a large community of this journal's readership. While I have provided a somewhat longwinded review below, I want to be clear that these comments are extremely minor – and mostly amount to a few minor points of clarification and extension that will increase readability for a larger audience.

We thank Reviewer 2 for the positive and thoughtful feedback.

2.2 Randomness parameter τ

1. In the introduction, the authors nicely clarify that randomness is different from overall exploratory bias (which is a different parameter). This point is sometimes confused in past literature but is indeed distinct as noted/modeled here. It would be helpful in the section on stochasticity to clarify that randomness can apply to (at least) three different metrics: (a) noise in whether to explore/exploit, (2) noise in where to exploit (if exploiting), and (3) noise in where to explore (if exploring). The randomness parameter τ in this paper captures random exploration (choice stochasticity in exploration location), which is appropriate for this task and similar to past studies from some of the authors of this group modeling this kind of task. This is just one minor clarification suggested for the introduction/overview; it was clear how this conceptual argument aligned with the analyses performed.

Thank you for raising this issue. To clarify, we have modified the following text to the introduction to clarify that other forms of randomness are also applicable:

Perhaps the most direct interpretation is to apply “cooling off” to the single dimension of random decision temperature, controlling the amount of noise when selecting actions or sampling hypotheses (Denison, Bonawitz, Gopnik, & Griffiths, 2013; Gopnik, Griffiths, & Lucas, 2015; Schulz et al., 2019), although alternative implementations are also possible (Dubois et al., 2020; Feng, Wang, Zarnescu, & Wilson, 2021).

2.3 Bayesian theories of development

2. Due to space limitations, I initially had trouble understanding how the point about changes in generalization and integration of new experiences differed conceptually from the following paragraph on Bayesian narrowing of priors. Providing a bit more space so the authors can clarify and expand on these points would go a long way – as these are complicated and abstract concepts that may be difficult for a less expert audience to grasp in the short-report space allotted. Specifically, it was initially unclear how both of these points aligned with the reported behavioral results, and only became more clear in the model-based analysis section. Generalization is defined here in terms of how observed reward values generalize to nearby unexplored spaces (e.g. the degree of similarity over distance assumed for reward distributions), but (as far as I can tell) is not also a parameter for learning this overall distance distribution over rounds. A brief point of clarification in the introduction re: type of generalization would go a long way. The Bayesian learning strategies are explored in terms of the stochasticity and rate of change of learning parameters (a kind of meta-level analysis). A few additional sentences of clarification and signposting in the introduction would also go a long way for a less expert reader.

We are grateful for the opportunity to clarify these important issues. First, we have added additional clarification about the connection between Bayesian narrowing of priors and stochastic optimization. We also expand in more detail on this connection in Section 2.5 in response to Reviewer 1, where we provide a graphical depiction of how Bayesian inference produces similar patterns of convergence. We quote below the changes we have made to the final paragraphs of the introduction:

Alternatively, one could apply the “cooling off” metaphor to an optimization process in the space of learning strategies, which can be characterized across multiple dimensions of learning. Development might thus be framed as parameter optimization, which tunes the parameters of an individual’s learning strategy, starting off by making large tweaks in childhood, followed by gradually lesser and more refined adjustments over the lifespan. In the metaphor, training iterations of the algorithm become a proxy for age.

This interpretation connects the metaphor of stochastic optimization with Bayesian models of cognitive development, which share a common notion of gradual convergence (Stamps & Frankenhuis, 2016; Tenenbaum et al., 2011). In Bayesian models of development, individuals in early developmental stages possess broad prior, and vague theories about the causal structure of the world, which become refined with experience (Tenenbaum et al., 2011). Bayesian principles dictate that over time, novel experiences will have a lesser impact on future beliefs or behavior as one’s priors become more narrow (Frankenhuis & Panchanathan, 2011; Stamps & Frankenhuis, 2016). Observed over the lifespan, this process will result in large changes to beliefs and behavior early in childhood and smaller changes in later stages, implying a similar developmental pattern as the stochastic optimization metaphor. Thus, not only might the outcomes of behavior be more stochastic during childhood, but the changes to the parameters governing behavior might also be more stochastic in earlier developmental stages.

Second, we have amended the text to clarify that by generalization we refer to reward generalization, from observed to novel choices. The following changes have been made at the first mention of generalization in the introduction:

Additionally, changes in how people generalize rewards to novel choices (Schulz et al., 2019) and the integration of new experiences (Blanco et al., 2016; Van den Bos et al., 2012) affect how beliefs are formed, and different actions are valued, also influencing choice variability.

Lastly, we appreciate the opportunity to clarify the distinction between our Bayesian reinforcement learning models and stochastic optimization applied as a meta-level analysis, explaining changes in the parameter estimates as a function of age. We now clarify this distinction in the “Goals and Scope” section:

We then provide direct empirical comparisons to multiple optimization algorithms as a meta-level analysis to describe changes in model parameters over the life span, where the best-performing algorithm is the most similar to human development.

2.4 Lack of motivation in young children

3. I also felt more discussion (and possibly additional analysis) could be given to two possible interpretations of this data. The first is that overall for ALL trials/decision making, children are following the explore-exploit decision expectations of the study and the children tend to show overall lower mean and max reward, higher unique options generated, a flattened search distance to reward, and decreased repeating (and increased “near clicks”). A second possibility is that younger children grow tired of the task within round (as they feel they have learned enough about the space), and meta-

goal shift to “play” with the squares, no longer attending to the explore-exploit optimization (potentially around trial 15-20 by examination of 2b). It’s possible that the explore-exploit behavior of children is quite similar to adults (though still arguably slower in their learning curves) up until this “pooping-out/play-goal-shifting point”, but that the decision to no longer engage in the original goals of the task leads to a big shift in behavior later in the task that thus dilutes overall mean and max reward, leads to higher unique options generated, flattened search distance, and decreased repeating (especially if the children are drawing pictures on the board or just clicking for fun). This idea is also related to the “more diverse strategies” point brought into the modeling section. It’s possible that overall children’s explore-exploit strategies (parameters) differed more from each other than adults overall, but it’s also possible that this result falls out because children had more varied interpretations of the overall meta-level task structure (and/or more goal shifting to meta “play” goals, etc within rounds) than adults, which led to this variability. To be clear, either explanation supports the broader points the authors are making in this manuscript and so should not be counted as a potential confound in interpretation. Nonetheless, it would be interesting to see further trial-by-trial analysis to investigate whether younger children hit a “poop-out” inflection point part way in (e.g. maybe splitting analysis for children around trial 17), to explore whether there is a behavioral shift that might partly explain the overall developmental results. My money is on the bet that the authors will find that overall (even in the first 16 trials) the results/parameters replicate, but that even stronger “random” behavior is observed in the later trials – suggesting that both overall children are more random and exploratory than adults, but also that they are less likely to stick with a task and “shift” at a goal/meta level part-way through each round that leads to even greater random and exploratory-bias. Nonetheless, I think it would be interesting to attempt to explore whether younger children are following task expectations (making explore-exploit decisions) throughout or whether they are switching up the task expectations all together and just playing or randomly clicking part way through because they lost interest in/value for the reward. This is a challenging analysis proposal because it is exploratory and post-hoc after examining the results – so please take this suggestion with a grain of salt. (Perhaps they make the most sense in a supplementary section.) Nonetheless, I think it is interesting to understand whether there are hints that children are strategy shifting within the task, or whether the results suggest consistent explore-exploit parameters of generalization, exploration bias, and randomness across all the trials.

We thank the reviewer for raising this interesting question. As suggested, we divided our analysis of the search distance by early (1-16) and late (17-25) trials (see Fig. R3). Whereas this plot shows major changes in adults’ search behavior (from sampling few repeat tiles in early trials to mostly sampling repeat tiles in late trials), we do not see similar shifts in the behavior of children. In early as well as late trials, they sample almost no tiles that they have chosen before, but rather sample near and to a smaller extent far options. If there was indeed a “poop-out” inflection point in children, we would expect to see more changes (e.g., convergence towards a random sampling strategy) in these search patterns as a function of trial.

In our supplementary analyses we also used a Bayesian hierarchical regression model to analyze if performance changes over rounds and did not find an effect of round and no reliable interaction between round and age group (see Fig. R4 reproduced below; Fig S2 and Table S2 in the supplementary section). Since we do not observe reliable changes to performance across rounds, we do not think there is evidence our results can be explained away simply due to boredom.

Figure R3: **Search decisions divided by early and late trials.** Proportion of repeat, near (distance=1) and far (distance > 1) choices divided by early and late trials. Each dot indicates a group mean, error bars indicate the 95% CI. Red dots indicate the random baseline.

2.5 Parameter names in the text

4. Please remind the readers what tau, lambda, and beta parameters capture whenever possible to help less technical readers. For example, “ λ and β parameters converged at lower values than the algorithm (...), while τ was not credibly higher or lower. . .” Could include light signposting to help with readability, e.g. “ λ (“generalization”) and β (“exploration bias”) parameters converged at lower values than the algorithm (...), while τ (“exploratory randomness”) was not credibly higher or lower...”.

Figure R4: **Learning over rounds.** Reward as a function of rounds. Each line is the fixed effect of a hierarchical Bayesian regression (Table S2) with the ribbons indicating 95% CI. Dots show the group mean

in each round and the red dashed line indicates a random baseline. We found no effect of round and no reliable interactions between round and age group.

We thank the reviewer for this suggestion. All mentions of model parameters in the main text have been signposted with the more intuitive, verbal description. Note that we now refer to τ as *decision* temperature whenever possible, to distinguish it from the *optimization* temperature of the stochastic optimization algorithms.

2.6 Diversity of participant sample

5. A stronger acknowledgement to the relative “WEIRD”ness of the sample, with implications for whether/how these findings could replicate with more representative world samples is needed. Both clarifying the sample characteristics and couching claims in terms of this narrowness are needed. I say this with the firm belief that the development of explore-exploit strategies examined here are likely to generalize across many populations, but also with the understanding that children who have experienced early childhood adversity may show differences in overall explore-exploit thresholds, with a preference for exploitation (Humphreys et al., 2015; Lloyd, McKay, & Furl, 2022). A nod to this work (and need for future work to explore and extend to Socio-economically, racially, globally, and early-stress-experience diverse populations is needed.

We thank the reviewer for pointing out this important limitation of our participant sample. We now address WEIRD bias and potential differences we might observe in more diverse cultural populations. This also provides a better transition to discussing the role of environmental mismatches in promoting maladaptive psychopathologies in the subsequent paragraph, where we integrated the two helpful reference suggestions:

Additionally, our participant sample is potentially limited by a WEIRD bias (Henrich, Heine, & Norenzayan, 2010), where the relative safety of developmental environments may promote more exploration. While we would expect to find a similar qualitative pattern in more diverse cultural settings, different expectations about the richness or predictability of environments may promote quantitatively different levels of each parameter. Indeed, previous work using a similar task but with risky outcomes found evidence for a similar generalization mechanism but a different exploration strategy that prioritized safety (Schulz et al., 2018).

Finally, our research also has implications for the role of the environment in maladaptive development. Our comparison to stochastic optimization suggests, in line with life history theory (Del Giudice, Gangestad, & Kaplan, 2016) and empirical work in rodents and humans (Dahl, Allen, Wilbrecht, & Suleiman, 2018; Gopnik et al., 2017; Lin et al., 2022), that childhood and adolescence are sensitive periods for configuring learning and exploration parameters. Indeed, adverse childhood experiences have been shown to reduce exploration and impair reward learning (Lloyd, McKay, & Furl, 2022). Organisms utilize early life experience to configure strategies for interacting with their environment, which for most species remain stable throughout the life span (Frankenhuis, Panchanathan, & Nettle, 2016). Once the configuration of learning strategies has cooled off, there is less flexibility for adapting to novel circumstances in later developmental stages. In the machine learning analogy we have used, some childhood experiences can produce a mismatch between training and test environments, where deviations from the expected environment have been linked to a number of psychopathologies (Humphreys & Zeanah, 2015).

2.7 Summary

I commend the authors on an excellent manuscript and hope they find these minor suggestions helpful

in future revisions.

We are very grateful for these helpful suggestions and for the positive feedback.

3 Reviewer 3

3.1 General remarks

The authors report analysis of an aggregated behavioural dataset in which children, adolescents and adults perform an RL task that requires spatial generalisation. Their question is how exploration (characterised here as a tendency to switch away from previously successful action) varies across the developmental trajectory. They report a number of interesting findings, and in particular (using a modelling approach) that exploration declines and then plateaus in the early teenage years, and that parameter fits become more similar across individuals with age. The former applies both to exploration modelled as a temperature parameter (general randomness) and the uncertainty-weight the computation of value (in a UCB model). The authors interpret their data as consistent with theories that have emphasised “cooling off” during development, with policies become increasingly crystallised over the lifespan.

I thought this was an impressive paper. I learned from it and would cite it. The work is clearly explained and nicely presented. The modelling is accomplished. I was initially worried about whether the authors model was identifiable, but the model / parameter recovery in the supplementary is convincing. It is nice to see lesion studies to show that all three parameters are needed.

We thank Reviewer 3 for careful examination of the model results and the positive assessment of the manuscript.

3.2 Reliability of model results

The major thought I had as reading the paper is whether the authors’ modelling results would be different if there were a simple improvement in performance across the lifespan due to generic factors that have nothing to do with exploration (for example, perhaps kids that are > 10 years simply understand the instructions better, or have a longer attention span, or better motor coordination). In particular given that tau and beta follow a similar trajectory it seems possible (although perhaps unlikely due to recovery success) that they are both tapping into a generic decrease in randomness at around 10 years. Here are some suggestions for how the authors could perhaps prevent similar worries in other readers.

We thank the reviewer for this important feedback. This is indeed a theme we have thought about in-depth in preparing this manuscript.

Firstly, we worked hard to ensure that participants of all ages could understand the experiment instructions. All participants started with an interactive tutorial (facilitated by a research assistant for children) in order to understand the dynamics of the task. Then, they were required to complete a number of comprehension checks before being allowed to start the task. While we cannot eliminate all doubt regarding potential differences in how well different ages understood the task, our procedure minimized this effect as much as possible.

Secondly, to test if there are differences in motivation over the course of the experiment, we conducted an analysis of performance as a function of round (Fig. S2; reproduced below as Fig. R5). This revealed no effect of round number on performance and no reliable interactions between age groups and rounds (Table S2). Thus, if there were age-related differences in motivation, we would expect this to be observable in this

analysis.

Figure R5: **Learning over rounds.** Reward as a function of rounds. Each line denotes the fixed effect of a hierarchical Bayesian regression (Table S2), with the ribbons indicating 95% CI. Dots show the group means in each round, and the red dashed line indicates a random baseline. We found no effect of round and no reliable interactions between round and age groups.

Lastly, our model is able to reliably recover separate and distinct parameters for uncertainty-directed exploration β and random decision temperature τ (Fig. S5; reproduced below as Fig. R6). This is evidence that these two parameters do not capture the same generic decrease in randomness around age 10. And while we cannot distinguish between sources of randomness in the decision temperature parameter τ (e.g., motor noise or more deliberate choice stochasticity), we can clearly distinguish this from a more sophisticated, directed exploration strategy captured by β .

Figure R6: **Parameter recovery (GP-UCB).** **a)** Parameter recovery using participant parameter estimates to generate data. The black dots denote individual parameter estimates. The colored lines show linear fit, while the dotted line denotes a diagonal indicative of perfect recovery. **b)** Augmented parameter recovery, using participant parameters but with systematic variation across a log-space grid of plausible counterfactual parameter values. This provides additional robustness by simulating data across a wider range of possible parameters. Black dots denote each simulated and generating parameter, while the colored dots denote the mean 95%CI. The colored lines show a linear fit, while the dotted line denotes a perfect recovery. The insetted statistics refer to the rank correlation between generative and fitted parameters, Kendall's τ .

3.3 Possible approaches

3.3.1 Modified regression models

1) could the authors perhaps compare models that involve restrictions on the change point separately for the three parameters? If I've understood correctly, here, the authors ask whether a model in which there is a single change point for all three parameters fares better than models with no or two or three change points and presumably flat (intercept only) models. However, could the data be better fit by a model in which only tau is changing (either linearly or with a single change point), but beta and lambda are not? This "restriction analysis" is similar to the lesion analysis but does not ask whether each parameter is required at all, but asks whether each parameter is needed to account for developmental change. This would potentially allow the authors to rule out another model in which there is a single generic factor that changes over development rather than to the more nuanced model that the authors favor.

This is a valid concern, and our conclusions would indeed have stronger support if we also show that all parameters are subject to a change point. We have now implemented a new model comparison that compares all possible configurations of either changepoint or intercept-only models for the three

parameters, using the same leave-one-out cross-validation procedure as before. These results reveal that age-related trends are still best described with changepoints in all parameters. We included the results of this model comparison procedure in Table S6 and have amended the text to the following:

Using leave-one-out cross-validation, we established that this simple changepoint model predicted all GP-UCB parameters better than linear or complex regression models up to fourth-degree polynomials (both with and without log-transformed variables and compared against lesioned intercept-only variants; see Tables S5-S6)

We are thankful for your help in making our conclusions more robust.

3.3.2 Relationship between exploration and performance

2) another route would be to say more about the relationship between exploration and performance. Simulated annealing has a normative explanation, as it's better to explore early and exploit later. But in this study, a tendency to exploit a known source of reward seems to be the reason why adults perform better, which makes the reader feel that perhaps younger participants are just generically doing worse, rather than strategically exploring more. Could the authors show – perhaps capitalising on individual differences but keeping age as a covariate – that there are *benefits* of exploration and that these benefits independently vary with age? For example, how does search distance early in play predict reward harvesting later, and how does this vary between individuals?

We thank the reviewer for raising this very important point. Disentangling the effects of exploration on performance is difficult, because not only does exploration influence performance, but performance influences exploration as well. For an example, more exploration can lead to finding higher rewards, while simultaneously, finding high rewards might reduce one's reliance on exploration by engaging more exploitation instead. Thus, we likely observe a bi-directional relationship between exploration and performance, which makes it difficult to establish causality.

Figure R7: **Effect of exploration on performance.** Normalized mean reward as a function of unique options sampled. Each dot indicates a participant mean, colored lines are the predictions of a linear model with 95% CI.

One attempt to analyze how the amount of exploration (number of unique options sampled) influences performance can be seen in Fig. R7. This plot shows a shift in the benefits of exploration: while young children (age 5-6) obtain higher rewards when exploring more, the obtained reward of the 7-8 year olds is independent of the amount of unique options they sample. For all the other age groups, performance decreases with an increasing amount of exploration. However, it's unclear which causal direction these results point to. Although there appears to be a benefit for young children who explore more, adults may also benefit from more exploration, but are quickly to switch to exploitation, thus reversing the trend. And while these results are intriguing, the issues with interpreting causality make us hesitant to include this analysis in the manuscript for fear of misinterpretation. However, future work could develop more targeted experiments to address this question, by manipulating the motivation for exploration within- participants.

3.3.3 Next-to strategy of children

3) It's striking to me that the youngest participants are more likely than adults to use a "next to" strategy (search distance = 1) but not a random spatial sampling strategy. Does the model recreate this finding? If not, what might be needed for it to do so? [minor: what is the dotted red line in Fig 2e? it's not referred to in the legend]. Related to point (2) above, are there benefits of this strategy? Do these benefits vary with age?

Past work with adult data has tried to account for this pattern of preferring neighboring options by adding a locality bias, which simply re-weights choices as a function of their distance from the previous choice

(Wu, Schulz, Speekenbrink, Nelson, & Meder, 2018). While this somewhat recreates the pattern of search distances, it does so without any clear interpretability, and yields the same pattern of parameter estimates (see Table S3: Wu et al., 2018). Conceptually, this imposes an external search cost, similar to how other paradigms impose a “stickiness” parameter for data with repeat choices being common.

While adding a locality bias can control for such patterns of behavior, they do not necessarily explain where they come from. A preference for local options can be due to a number of different factors, which include both cognitive (e.g., originating from adaptive foraging behavior; Hills, Brockie, & Maricq, 2004; Kareiva & Odell, 1987) and non-cognitive sources (e.g., search costs or amplified motor noise when moving to more distant options; Fitts & Peterson, 1964).

We can speculate about the potential cognitive benefits of local search, since locality prioritizes exploring a locally coherent region of the search space. This might provide potential advantages for learning about the structure of the environment. Our past work shows we can control for it, with the same overall pattern of model parameters (Wu et al., 2018). Yet, it is still an open question for future research to distinguish between cognitive and motoric sources of locality. However, the current paradigm is not designed to adequately dissociate between the cognitive and motoric sources of locality.

We thank the reviewer for raising the issue of interpreting the red line in Fig. 2e. As in all other figures, it provides a comparison to a random baseline and is defined in the panel a caption. We have added additional emphasis in the figure caption.

3.4 Summary

So in short I think this is a great paper but I would love it if the authors went a bit further in showing that young people’s behaviour is “exploration” and not just “error”.

We are extremely grateful for this positive evaluation of our manuscript and appreciate the thoughtful concerns presented in the review. We think this issue of higher error instead of exploration in younger participants is extremely important and relates to current debates about how best to interpret parameters obtained from fitting a cognitive model to behavior (Eckstein et al., 2022; Eckstein, Wilbrecht, & Collins, 2021; Navarro, 2019). While this debate is unfortunately outside of the scope of this manuscript, here, we have strong reason to believe potential issues of misidentification are minimal if at all present.

First, the GP-UCB model has two components that capture independent and recoverable aspects of exploration (figure R6). β captures uncertainty-directed exploration while decision temperature τ captures random exploration, the latter being one possible implementation of random error. Younger individuals have higher estimates in both β and τ , which makes us confident there is more than random error to even the youngest participants’ behavior. Indeed, the simulated learning curves in Figure 3d suggest that without accounting for uncertainty-directed exploration (β -lesion model), we fail to capture the speed of learning in older participants and the decaying reward curves in younger children. Thus, this pattern of behavior cannot be captured by random error alone (i.e., τ).

Second, the τ -lesion model provides an alternative (and potentially more direct) implementation of random “error” (Zilker, 2022). While we indeed see in Figure 3a that the τ -lesion model has some probability of being the best model in the youngest participants, the GP-UCB model’s posterior probability strikingly exceeds that of the τ -lesion model. The simulated learning curves of the τ -lesion model in Figure 3d also fail to capture qualitative patterns of behavior.

Third, all participants perform reliably better and display distinctly different search patterns compared to a random model (red lines in Figure 2). Thus, if young people’s exploration is solely error, we would expect their behavioral patterns to resemble the random model.

We thus hope to convince the reviewer, in both our response and additional analyses to the previous sections, that the exploration behavior we capture is not just error. We are very grateful for these thoughtful

comments, which have improved the manuscript substantially.

References

- Blanco, N. J., Love, B. C., Ramscar, M., Otto, A. R., Smayda, K., & Maddox, W. T. (2016). Exploratory decision-making as a function of lifelong experience, not cognitive decline. *Journal of Experimental Psychology: General*, *145*(3), 284.
- Blanco, N. J., & Sloutsky, V. M. (2021). Systematic exploration and uncertainty dominate young children's choices. *Developmental Science*, *24*(2), e13026.
- Dahl, R. E., Allen, N. B., Wilbrecht, L., & Suleiman, A. B. (2018, February). Importance of investing in adolescence from a developmental science perspective. *Nature*, *554*(7693), 441–450.
- Del Giudice, M., Gangestad, S. W., & Kaplan, H. S. (2016). Life history theory and evolutionary psychology.
- Denison, S., Bonawitz, E., Gopnik, A., & Griffiths, T. L. (2013). Rational variability in children's causal inferences: The sampling hypothesis. *Cognition*, *126*(2), 285–300.
- Dubois, M., Aislinn, B., Moses-Payne, M. E., Habicht, J., Steinbeis, N., & Hauser, T. U. (2020). Tabulara exploration decreases during youth and is linked to adhd symptoms. *BioRxiv*.
- Eckstein, M. K., Master, S. L., Xia, L., Dahl, R. E., Wilbrecht, L., & Collins, A. G. E. (2022, November). The interpretation of computational model parameters depends on the context. *Elife*, *11*.
- Eckstein, M. K., Wilbrecht, L., & Collins, A. G. E. (2021, October). What do reinforcement learning models measure? interpreting model parameters in cognition and neuroscience. *Curr Opin Behav Sci*, *41*, 128–137.
- Feng, S. F., Wang, S., Zarnescu, S., & Wilson, R. C. (2021). The dynamics of explore–exploit decisions reveal a signal-to-noise mechanism for random exploration. *Scientific reports*, *11*(1), 1–15.
- Fitts, P. M., & Peterson, J. R. (1964, February). Information capacity of discrete motor responses. *Journal of Experimental Psychology*, *67*, 103–112.
- Frankenhuis, W. E., & Panchanathan, K. (2011, December). Balancing sampling and specialization: an adaptationist model of incremental development. *Proc. Biol. Sci.*, *278*(1724), 3558–3565.
- Frankenhuis, W. E., Panchanathan, K., & Nettle, D. (2016). Cognition in harsh and unpredictable environments. *Current Opinion in*.
- Gerstenberg, T., & Tenenbaum, J. B. (2017). Intuitive Theories. In *The Oxford Handbook of Causal Reasoning*. Oxford University Press. doi: 10.1093/oxfordhb/9780199399550.013.28
- Gopnik, A., Griffiths, T. L., & Lucas, C. G. (2015). When younger learners can be better (or at least more open-minded) than older ones. *Current Directions in Psychological Science*, *24*(2), 87–92.
- Gopnik, A., O'Grady, S., Lucas, C. G., Griffiths, T. L., Wente, A., Bridgers, S., ... Dahl, R. E. (2017). Changes in cognitive flexibility and hypothesis search across human life history from childhood to adolescence to adulthood. *Proceedings of the National Academy of Sciences*, *114*(30), 7892–7899.
- Henrich, J., Heine, S. J., & Norenzayan, A. (2010). Most people are not weird. *Nature*, *466*(7302), 29–29.
- Hills, T. T., Brockie, P. J., & Maricq, A. V. (2004, February). Dopamine and glutamate control area-restricted search behavior in *caenorhabditis elegans*. *J. Neurosci.*, *24*(5), 1217–1225.
- Humphreys, K. L., & Zeanah, C. H. (2015). Deviations from the expectable environment in early childhood and emerging psychopathology. *Neuropsychopharmacology*, *40*(1), 154–170.
- Kareiva, P., & Odell, G. (1987, August). Swarms of predators exhibit “preytaxis” if individual predators use area-restricted search. *Am. Nat.*, *130*(2), 233–270.
- Lin, W. C., Liu, C., Kosillo, P., Tai, L.-H., Galarce, E., Bateup, H. S., ... Wilbrecht, L. (2022, July). Transient food insecurity during the juvenile-adolescent period affects adult weight, cognitive flexibility, and dopamine neurobiology. *Curr. Biol.*

- Lloyd, A., McKay, R. T., & Furl, N. (2022). Individuals with adverse childhood experiences explore less and underweight reward feedback. *Proceedings of the National Academy of Sciences*, 119(4), e2109373119.
- Meder, B., Wu, C. M., Schulz, E., & Ruggeri, A. (2021). Development of directed and random exploration in children. *Developmental Science*, e13095. doi: 10.1111/desc.13095
- Navarro, D. J. (2019, March). Between the devil and the deep blue sea: Tensions between scientific judgement and statistical model selection. *Computational Brain & Behavior*, 2(1), 28–34.
- Rule, J. S., Tenenbaum, J. B., & Piantadosi, S. T. (2020). The child as hacker. *Trends in cognitive sciences*, 24(11), 900–915.
- Schulz, E., Wu, C. M., Huys, Q. J., Krause, A., & Speekenbrink, M. (2018). Generalization and search in risky environments. *Cognitive Science*, 42, 2592–2620. doi: 10.1111/cogs.12695
- Schulz, E., Wu, C. M., Ruggeri, A., & Meder, B. (2019). Searching for rewards like a child means less generalization and more directed exploration. *Psychological Science*, 30(11), 1561–1572. doi: 10.1177/0956797619863663
- Somerville, L. H., Sasse, S. F., Garrad, M. C., Drysdale, A. T., Abi Akar, N., Insel, C., & Wilson, R. C. (2017). Charting the expansion of strategic exploratory behavior during adolescence. *Journal of experimental psychology: general*, 146(2), 155.
- Stamps, J. A., & Frankenhuis, W. E. (2016, April). Bayesian models of development. *Trends Ecol. Evol.*, 31(4), 260–268.
- Tenenbaum, J. B., Kemp, C., Griffiths, T. L., & Goodman, N. D. (2011, March). How to grow a mind: statistics, structure, and abstraction. *Science*, 331(6022), 1279–1285.
- Van den Bos, W., Cohen, M. X., Kahnt, T., & Crone, E. A. (2012). Striatum–medial prefrontal cortex connectivity predicts developmental changes in reinforcement learning. *Cerebral cortex*, 22(6), 1247–1255.
- Wu, C. M., Schulz, E., Garvert, M. M., Meder, B., & Schuck, N. W. (2020, September). Similarities and differences in spatial and non-spatial cognitive maps. *PLOS Computational Biology*, 16, 1–28. doi: 10.1371/journal.pcbi.1008149
- Wu, C. M., Schulz, E., & Gershman, S. J. (2021). Inference and search on graph-structured spaces. *Computational Brain & Behavior*, 4, 125–147. doi: 10.1007/s42113-020-00091-x
- Wu, C. M., Schulz, E., Pleskac, T. J., & Speekenbrink, M. (2022). Time pressure changes how people explore and respond to uncertainty. *Scientific Reports*, 12, 1–14. doi: <https://doi.org/10.1038/s41598-022-07901-1>
- Wu, C. M., Schulz, E., Speekenbrink, M., Nelson, J. D., & Meder, B. (2018). Generalization guides human exploration in vast decision spaces. *Nature Human Behaviour*, 2, 915–924. doi: 10.1038/s41562-018-0467-4
- Zilker, V. (2022, September). Choice rules can affect the informativeness of model comparisons. *Computational Brain & Behavior*, 5(3), 397–421.

Decision Letter, first revision:

18th May 2023

Dear Dr. Wu,

Thank you for your patience as we've prepared the guidelines for final submission of your Nature Human Behaviour manuscript, "Developmental changes resemble stochastic optimization" (NATHUMBEHAV-22112969A). Please carefully follow the step-by-step instructions provided in the attached file, and add a response in each row of the table to indicate the changes that you have

made. Please also address the additional marked-up edits we have proposed within the reporting summary. Ensuring that each point is addressed will help to ensure that your revised manuscript can be swiftly handed over to our production team.

We would hope to receive your revised paper, with all of the requested files and forms within two-three weeks. Please get in contact with us if you anticipate delays.

Nature Human Behaviour offers a Transparent Peer Review option for new original research manuscripts submitted after December 1st, 2019. As part of this initiative, we encourage our authors to support increased transparency into the peer review process by agreeing to have the reviewer comments, author rebuttal letters, and editorial decision letters published as a Supplementary item. When you submit your final files please clearly state in your cover letter whether or not you would like to participate in this initiative. Please note that failure to state your preference will result in delays in accepting your manuscript for publication.

In recognition of the time and expertise our reviewers provide to Nature Human Behaviour's editorial process, we would like to formally acknowledge their contribution to the external peer review of your manuscript entitled "Developmental changes resemble stochastic optimization". For those reviewers who give their assent, we will be publishing their names alongside the published article.

Cover suggestions

As you prepare your final files we encourage you to consider whether you have any images or illustrations that may be appropriate for use on the cover of Nature Human Behaviour.

ORCID

Non-corresponding authors do not have to link their ORCIDs but are encouraged to do so. Please note that it will not be possible to add/modify ORCIDs at proof. Thus, please let your co-authors know that if they wish to have their ORCID added to the paper they must follow the procedure described in the following link prior to acceptance: <https://www.springernature.com/gp/researchers/orcid/orcid-for-nature-research>

Nature Human Behaviour has now transitioned to a unified Rights Collection system which will allow our Author Services team to quickly and easily collect the rights and permissions required to publish your work. Approximately 10 days after your paper is formally accepted, you will receive an email in providing you with a link to complete the grant of rights. If your paper is eligible for Open Access, our Author Services team will also be in touch regarding any additional information that may be required to arrange payment for your article. Please note that you will not receive your proofs until the publishing agreement has been received through our system.

Please note that *Nature Human Behaviour* is a Transformative Journal (TJ). Authors may publish their research with us through the traditional subscription access route or make their paper immediately open access through payment of an article-processing charge (APC). Authors will not be required to make a final decision about access to their article until it has been accepted. Find out more about Transformative Journals

[REDACTED]

Best regards,
Alex McKay
Editorial Assistant
Nature Human Behaviour

On behalf of

Jamie

Dr Jamie Horder
Senior Editor
Nature Human Behaviour

Reviewer #2:
Remarks to the Author:

The authors have provided a thoughtful and thorough response to the reviews. What was already a very strong paper is now even stronger. I have no further suggestion and commend the authors on a lovely contribution to the field.

Reviewer #3:
Remarks to the Author:

Thanks for a responsive revision. I don't have any further concerns.

Author Rebuttal, first revision:

Decision Letter, second revision:

Final Decision Letter:

Dear Dr Wu,

We are pleased to inform you that your Article "Developmental changes in exploration resemble stochastic optimization", has now been accepted for publication in *Nature Human Behaviour*.

Please note that *Nature Human Behaviour* is a Transformative Journal (TJ). Authors whose manuscript was submitted on or after January 1st, 2021, may publish their research with us through the traditional subscription access route or make their paper immediately open access through payment of an article-processing charge (APC). Authors will not be required to make a final decision about access to their article until it has been accepted. IMPORTANT NOTE: Articles submitted before January 1st, 2021, are not eligible for Open Access publication. Find out more about Transformative Journals

Acceptance of your manuscript is conditional on all authors' agreement with our publication policies (see <http://www.nature.com/nathumbehav/info/gta>). In particular your manuscript must not be

published elsewhere and there must be no announcement of the work to any media outlet until the publication date (the day on which it is uploaded onto our web site).

With best regards,

Jamie

Dr Jamie Horder
Senior Editor
Nature Human Behaviour

P.S. Click on the following link if you would like to recommend Nature Human Behaviour to your librarian <http://www.nature.com/subscriptions/recommend.html#forms>

** Visit the Springer Nature Editorial and Publishing website at www.springernature.com/editorial-and-publishing-jobs for more information about our career opportunities. If you have any questions please click here.**

This email has been sent through the Springer Nature Tracking System NY-610A-NPG&MTS

Confidentiality Statement:

This e-mail is confidential and subject to copyright. Any unauthorised use or disclosure of its contents is prohibited. If you have received this email in error please notify our Manuscript Tracking System Helpdesk team at <http://platformsupport.nature.com>.

Details of the confidentiality and pre-publicity policy may be found here <http://www.nature.com/authors/policies/confidentiality.html>